



**Carbon mineralization in Laptev and East Siberian Sea shelf and slope sediment**
Volker Brüchert[1,3], Lisa Bröder[2,3], Joanna E. Sawicka[1,3], Tommaso Tesi[2,3,5], Samantha P.
Joye[6], Xiaole Sun[4,5], Igor P. Semiletov[7,8,9], Vladimir A. Samarkin[6]
[1] Department of Geological Sciences, Stockholm University, Stockholm, Sweden
[2] Department of Environmental Sciences and Analytical Chemistry, Stockholm University,
Stockholm, Sweden
[3] Bolin Centre for Climate Research, Stockholm University, Stockholm, Sweden
[4] Baltic Sea Research Center, Stockholm University, Stockholm, Sweden
[5] Institute of Marine Sciences – National Research Council, Bologna, Italy
[6] Department of Marine Sciences, University of Georgia, Athens, U.S.A.
[7] International Arctic Research Center, University Alaska Fairbanks, Fairbanks, USA
[8] Pacific Oceanological Institute, Russian Academy of Sciences, Vladivostok, Russia
[9] Tomsk National Research Politechnical University, Tomsk, Russia
**Abstract** The Siberian Arctic Sea shelf and slope is a key region for the degradation of
terrestrial organic material transported from the organic carbon-rich permafrost regions of
Siberia. We report on sediment carbon mineralization rates based on $O_2$ microelectrode
profiling, intact sediment core incubations, $^{35}$S-sulfate tracer experiments, porewater
dissolved inorganic carbon (DIC), $\delta^{13}C_{DIC}$, and iron, manganese, and ammonium
concentrations from 20 shelf and slope stations. This data set provides a spatial overview of
sediment carbon mineralization rates and pathways over large parts of the outer Laptev and
East Siberian Arctic shelf and slope, and allowed us to assess degradation rates and efficiency
of carbon burial in these sediments. Rates of oxygen uptake and iron and manganese
reduction were comparable to temperate shelf and slope environments, but bacterial sulfate
reduction rates were comparatively low. In the topmost 20 to 50 cm of sediment, aerobic
carbon mineralization dominated degradation and comprised on average 82% of the depth-
integrated carbon mineralization. Oxygen uptake rates and $^{35}$S-sulfate reduction rates were
higher in the eastern East Siberian Sea shelf compared to the Laptev Sea shelf. DIC/NH$_4^+$
ratios in porewaters and the stable carbon isotope composition of remineralized DIC indicated
that the degraded organic matter on the Siberian shelf and slope was a mixture of marine and



terrestrial organic matter. Based on dual end member calculations, the terrestrial organic
carbon contribution varied between 32% and 36%, with a higher contribution in the Laptev
Sea than in the East Siberian Sea. Extrapolation of the measured degradation rates using
isotope end member apportionment over the outer shelf of the Laptev and East Siberian Sea
suggests that about 16 Tg C per year are respired in the outer shelf sea floor sediment. Of the
organic matter buried below the oxygen penetration depth, between 0.6 and 1.3 Tg C per year
are degraded by anaerobic processes, with a terrestrial organic carbon contribution ranging
between 0.3 and 0.5 Tg per year.
Key words: Carbon mineralization, Arctic shelf and slope sediment, Laptev Sea, East Siberian
Sea
**1. Introduction**
The biogeochemical fate of terrestrial organic carbon deposited on the Arctic shelf and
slope is one of the most important open questions for the marine Arctic carbon cycle (e.g.,
Tesi et al., 2014; Macdonald et al., 2015; McGuire et al., 2009; Vonk et al, 2012). The total
pan-Arctic terrestrial permafrost carbon reservoir has been estimated at about 1100 – 1500 Pg
(Hugelius et al, 2014) – a carbon pool large enough to substantially affect the global
atmospheric carbon dioxide pool over the next 100 years, even when only partially
decomposed after thawing and oxidation (Schuur et al. 2015; Koven et al., 2015). Yet, there
remains considerable uncertainty regarding the mineralization of terrestrial organic matter
exported by rivers and coastal erosion to the Siberian shelf and slope (Tesi et al., 2014;
Karlsson et al. 2015; Semiletov et al., 2011; Salvado et al., 2016).
Terrestrial organic matter transported to the Siberian shelf is of variable size, age, and
molecular composition, which results in a range of qualitatively different carbon degradation
rates of bulk carbon and individual molecular components. Size class analysis of the organic
matter suggests that coarse organic material settles preferentially in near-shore environments,
whereas finer organic fractions disperse offshore in repeated deposition-resuspension cycles
gradually losing particular molecular components and overall reactivity (Wegner et al., 2013;
Tesi et al., 2014, 2016). Substantial oxic degradation of organic matter may occur during
near-bottom transport in resuspension-deposition cycles across the shelf (Bröder et al.,
2016a). Up to 90% of certain biomarker classes may decompose during transport, whereby



most of the degradation may take place while the transported organic material resides in the
sediment before being resuspended (Bröder et al., 2016a). However, without making
approximations on transport direction, particle travel time and travel distance these studies
cannot provide direct insights into the rates of carbon degradation and resultant $CO_2$ fluxes
from sediment. By contrast, direct kinetic constraints provided by sediment carbon
degradation rates can provide testable data for coupled hydrodynamic biogeochemical models
that help assess the fate of land-exported terrestrial carbon pool on the Siberian shelf.

Relatively few studies have directly measured rates of carbon mineralization rates in

Siberian shelf sediment (e.g., Boetius and Damm, 1998; Grebmeier et al., 2006; Karlsson et
al., 2015, Savvichev et al., 2007). Boetius and Damm (1998) used high-resolution oxygen
microelectrode data to determine the surface oxygen concentration gradients and oxygen
penetration depths in a large number of sediment cores from the shelf and slope of the Laptev
Sea. Based on corresponding sediment trap and export productivity data, they concluded that
the annual marine organic carbon export in the Laptev Sea shelf and slope was sufficiently
high to explain the observed oxygen uptake rates. Current understanding therefore holds that
due to the long annual ice cover and low productivity on the eastern Siberian Arctic shelf and
slope, only a small amount of marine organic carbon is exported and buried in Laptev and
East Siberian Sea shelf sediment. The highly reactive fraction of fresh organic matter is
thought to degrade in the surface sediment (Boetius and Damm, 1998). Consequently,
anaerobic respiration in buried sediment has been thought to be negligible and to reflect the
degradation of unreactive terrestrially derived carbon compounds. To our knowledge, with the
exception of a recent study by Karlsson et al (2015) a more direct assessment of terrestrial
carbon-derived mineralization rates in buried shelf and slope sediment has not been reported
for the East Siberian Arctic Sea.

In this study, we present data from oxygen microelectrode profiling experiments,

porewater data of dissolved inorganic carbon and its stable carbon isotope composition, and
$^{35}$S-sulfate reduction rate experiments along a shelf-slope transect near 125°E in the Laptev
Sea. Samples were taken during the summer 2014 on the SWERUS-C3 expedition with the
Swedish icebreaker Oden. We combined these data with porewater analysis of dissolved
ammonium, sulfate, iron, and manganese to assess the major carbon degradation pathways
and rates across the extensive outer Laptev and Siberian shelf and slope.





## 2. Materials and methods

### 2.1. Sample collection

Samples were collected at 20 stations from 40 to 3146 m water depth in the western Laptev and East Siberian Sea (Fig. 1 and Table 1). In this study we only report on sampling sites that showed no methane gas plumes, acoustic anomalies in the water column, or sediment blankings indicative of rising gas. In areas of active ebullition from the seafloor as seen by video imagery and acoustic gas blankings in the water column, the biogeochemistry of sea floor processes such as bacterial sulfate reduction, DIC concentration and its carbon isotope composition, and oxygen uptake are affected by methane oxidation. These methane cycling-related signals overprint the biogeochemistry imparted by carbon mineralization and are reported in a separate study.

Sediment stations had variable ice cover (Table 1). In the Laptev Sea, except for the deep-water slope stations between 3146 m and 2106m, all stations had open water. By contrast, ice cover exceeded 75% in the East Siberian Sea to the west and east of Bennett island (Station 40 to 63). Sediments with well-preserved sediment surfaces were collected with a Multicorer (Oktopus GmbH, Kiel, Germany) that simultaneously takes 8 sediment cores over an area of about 0.36 $m^2$ with acrylic tubes (9.5 cm diameter, 60 cm length) to 40 cm depth preserving clear water on top of the sediment. At stations 6, 23, and 24 an underwater video system (Group B Distribution Inc., Jensen Beach, U.S.A.) was mounted on the multicore frame to record the deployment and recovery, and to document the sea floor habitat. For the investigations all cores were taken from the same cast. Two of the cores were used to determine $^{35}$S-sulfate reduction rates and porosity. In addition, one core with predrilled 3.8 mm holes sealed with electric tape was used to extract porewaters with rhizons (Rhizosphere Research Products BV, Wageningen, Netherlands). A fifth core was used for microelectrode measurements of dissolved oxygen concentration profiling, and finally, four other cores were used for whole-core incubations to determine benthic fluxes of dissolved oxygen, dissolved inorganic carbon, and nutrients. The cores were capped with rubber stoppers until further subsampling usually within 30 minutes. For sulfate reduction rates, the cores were subsampled with 40 or 50 cm long acrylic tubes (26 mm inner diameter) prepared with silicon-sealed holes, drilled at distances of 1 cm. For whole-core incubations, the cores were sub-sampled with 25 cm-long, 60 mm-wide tubes (56 mm id) to 12 cm depth. Likewise, a 60 mm diameter tube (56 mm id) was collected for microelectrode measurements preserving



about 3 cm of the overlying bottom water. For intact whole-core incubations, temperature-
controlled aquaria were filled with bottom water that was collected from a CTD rosette from
the same station by collecting water from four ten-liter rosette bottles usually ~5 meters above
the sea floor. All sediment cores were closed with a stopper retaining the water on top of the
sediment and stored at 1.5°C in an incubator until further processing.

## 2.2.  Microelectrode oxygen profiles

High-resolution $O_2$-profiles across the water-sediment interface were obtained to
determine oxygen penetration depths and diffusive oxygen uptake (Rasmussen and Jørgensen,
1992; Glud, 2008). The 60 mm tubes were placed in an aquarium filled with bottom water
from the same station, overflowing the sediment core. The water temperature was kept to
~1°C by a cooling unit (Julabo GmbH, Seelbach, Germany). In exceptional cases when there
was not sufficient bottom water available to fill the aquarium, bottom water was used from a
pump system. A stable diffusive boundary layer above the sediment was created by passing
air from an aquarium pump over the water surface with a Pasteur pipette creating a slow
rotational motion of water inside the core. At each station six to eight $O_2$ microprofiles were
measured using Clark-type oxygen microelectrodes (OX-50, Unisense, Århus Denmark)
mounted on a motor-driven micromanipulator (MM33, Unisense, Århus Denmark). $O_2$
sensors were calibrated with fully oxygenated bottom water from the same station at ~1°C for
saturation and for anoxic conditions by dissolving $Na_2SO_3$ dissolved in the same water. The
first profile in each core was measured with a resolution of 1000µm as a quick scan to locate
the sediment surface and to adjust the measuring range. Then the vertical resolution was
increased to 100-500µm and additional five to seven profiles were measured at different
points on the surface, approximately one cm apart from each other.

## 2.3. Whole-core sediment incubations

Four intact cores with undisturbed sediment surfaces and clear overlying water were
subsampled in the laboratory in acrylic tubes (i.d. 56 mm, height 25 cm) retaining about 10
cm of the overlying water. The sediment and water height in the tubes were approximately 10
cm. The cores were incubated in a 40-liter incubation tank filled with bottom water from the
same station. Before the incubation the overlying water in the cores was equilibrated with




bottom water in the tank. The overlying water in the cores was stirred by small magnetic bars
mounted in the core liners and driven by an external magnet at 60 rpm. The cores were pre-
incubated uncapped for 6 hours and subsequently capped and incubated for a period of 6 to 24
hours depending on the initial oxygen concentration in the bottom water. 2D oxygen sensor
spots (Firesting oxygen optode, PyroScience GmbH, Aachen, Germany) with a sensing
surface of a diameter of 5 mm were attached to the inner wall of two incubation cores. The
sensor spots were calibrated against $O_2$-saturated bottom water and oxygen-free water
following the manufacturer's guidelines accounting for temperature and salinity of the
incubation water. Measurements were performed with a fiberoptic cable connected to the spot
adapter fixed at the outer core liner wall at the spot position. The $O_2$ concentration was
continuously logged during incubations. Sediment total oxygen uptake (TOU) rates were
computed by linear regression of the $O_2$ concentration over time. 5 ml of overlying water
were removed over the course of the incubation used for nutrient and dissolved $CO_2$ analysis
as described below. Linear regression best fits were used to determine the exchange fluxes of
dissolved $CO_2$ and ammonium.

## 2.4. Extracted porewater analysis

Porewater samples for concentration measurements of total dissolved $CO_2$ (DIC),
sulfate and ammonium were obtained using the methods described in Seeberg-Elverfeldt et al.
(2005). Rhizons were treated for 2 hours in 10% HCl solution, followed by two rinses with
deionized water for 2 hours and final storage in deionized water. The rhizons were connected
to 10 mL disposable plastic syringes with inert pistons (VWR, Stockholm, Sweden) via
polyethylene 3-way luer-type stopcocks (Cole-Parmer, U.S.A.) and inserted in 1 cm intervals
through tight-fitting, pre-drilled holes in the liner of the sediment cores. The first mL of pore
water was discarded from the syringe. No more than 5 ml were collected from each core to
prevent cross-contamination of adjacent porewater due to the suction effect (Seeberg-
Elverfeldt et al., 2005). The collected porewater was divided into four different aliquots for
later chemical analysis. For dissolved sulfate analysis, 1 ml of porewater was preserved with
200 µl of 5% zinc acetate solution and frozen. For ICP-AES analysis of dissolved metals and
major cations, 1 ml of porewater was preserved with 100 µl of 10% Suprapur $HNO_3$ and
stored cold. For analysis of dissolved ammonium, 2 ml of porewater were frozen untreated.
For analysis of dissolved inorganic carbon, 2 ml of porewater were preserved with 100 µl




10% $HgCl_2$ and stored cold in brown glass vials without headspace. Sulfate concentration was
measured on diluted aliquots on a Dionex System IC 20 ion chromatograph. DIC
concentrations were determined by flow injection analysis (Hall and Aller, 1992). Ammonium
was determined on a QUAATRO 4-channel flow injection analyzer (Seal Analytical).
Dissolved iron and manganese were determined on diluted aliquots by ICP-AES (Varian
Vista AX). For carbon isotope analysis of dissolved inorganic carbon, 1 ml of porewater was
filled into 12 ml exetainers to which 1 ml of concentrated phosphoric acid was added. The
carbon isotope composition of the formed $CO_2$ was analyzed on a GasbenchII-MAT 253
isotope ratio mass spectrometer coupled to a GC-PAL autosampler. Results are reported in the
conventional delta notation relative to PDB. Precision of isotope analysis is 0.1‰.

**2.5. Reaction transport modelling**
Reaction rates and fluxes were estimated from the concentration profiles of dissolved
oxygen, manganese, iron, and dissolved inorganic carbon according to the general reaction-
transport equation accounting for diffusion and advection exemplified here for dissolved
oxygen according to
$$\frac{dO_2}{dt} = \frac{\partial D_s \frac{\partial O_2}{\partial z}}{\partial z} + \frac{\partial v O_2}{\partial z} + \omega \frac{\partial O_2}{\partial z} + \sum R \qquad (1)$$
At steady state, the rate of the concentration change reflects the balance between the
consumption due to respiration and oxidation of reduced inorganic compounds (R) against
diffusion and advection due to bioirrigation into sediment (Glud, 2008). $D_s$ is the sediment
diffusion coefficient and was calculated for the experimental temperature and salinity
according to Boudreau (1997). The sediment diffusion coefficient $D_s$ was recalculated from
the molecular diffusion coefficient $D_o$ according to $D_s = D_o / \theta^2$, where $\theta^2 = 1 - \ln(\varphi^2)$, where
$\varphi$ is porosity and $\theta$ is tortuosity (Boudreau, 1997). Sediment burial $\omega$ was based on [210]Pb-
based sediment accumulation rates (Bröder et al., 2016b). The advection rate $\upsilon$ was estimated
by stepwise optimization by fitting an oxygen concentration profile to the measured
concentration data using the least square fitting procedure of the program Profile (Berg et al.,

1998).






### 2.6. $^{35}$S-Sulfate reduction rates


Bacterial sulfate reduction rates ($^{35}$S-SRR) sediment cores were subsampled in 40-cm
long 28 mm-diameter cores with 1-cm spaced, silicon-sealed, pre-drilled small holes on the
side for injections. For the incubation, the whole-core incubation method by Jørgensen (1978)
was used. $^{35}SO_4^{2-}$ tracer solution was diluted in a 6 ‰ NaCl solution containing 0.5 mM $SO_4^{2-}$
and 2.5 µl of the tracer solution (200kBq) was injected through the pre-drilled holes. The
cores were then capped and sealed in plastic wrap foil and incubated for 8 hours at the
respective bottom water temperatures. After this time, the incubations were stopped by
sectioning the core in 1-cm intervals to 5 cm depth and in two centimeter intervals below this
depth to the bottom of the core. Sediment sections were transferred into 50 ml plastic
centrifuge tubes containing 20 ml zinc acetate (20% v/v) and shaken vigorously and frozen.
The total amount of $^{35}$S-labeled reduced inorganic sulfur (TRS) was determined using the
single-step cold chromium distillation method by Kallmeyer et al. (2004). TRS and
supernatant sulfate were counted on a TriCarb 2095 Perkin Elmer scintillation counter. The
sulfate reduction rate was calculated using the following equation (Jørgensen, 1978):
$$^{35}SRR = \left( {TRI^{35}S \times 1.045} \middle/ {(^{35}SO_4^{2-} + TRI^{35}S)} \right) \times [SO_4^{2-}]/\rho T \qquad (2)$$
where [$SO_4^{2-}$] is the pore water sulfate concentration corrected for porosity $\rho$, TRI$^{35}$S and
$^{35}SO_4^{2-}$ are the measured counts (cpm) of sulfate and total reduced inorganic sulfur species,
respectively, 1.045 is a correction factor accounting for the kinetic isotope effect of $^{35}$S
relative to $^{32}$S, and T is the incubation time. The sulfate reduction rate is reported as nmol cm$^{-3}$
$^3$ day$^{-1}$. $^{35}$SRR were measured in two parallel cores for all depth intervals. The incubation
experiments were conducted between July 15 and August 20, but for logistical reasons
(transport to Stockholm) the distillation of the samples was conducted between December 10,
2014 and April 2, 2015 so that between 1.7 and 2.7 half-lives of $^{35}$S (87.4 days) had passed
before all samples were processed. The resulting detection limit of the rate measurements
accounting for distillation blanks and radioactive decay of $^{35}$S between experiment and
laboratory workup was 0.01 nmol cm$^{-3}$ day$^{-1}$.




## 3. Results

### 3.1. Physical and chemical bottom water conditions

Table 1 summarizes the general site characteristics of the investigated sediment stations. Bottom water salinity varied between 34.9 ‰ in the outer Laptev Sea at 3146 m depth (Station 1) to 29.1 ‰ in the East Siberian Sea at 40 m (Station 45). The lower salinity in the East Siberian Sea can be attributed to longshore transport of freshwater eastward from the Lena River. Bottom water temperatures varied between -1.8°C at Station 27 and 0°C at Station 37, but there was no regional trend in the data. Cored sediment consisted of silty clays to clayey silts. Slope sediment had a distinctly chocolate brown color throughout the cored interval, whereas shelf sediment only had a 1 to 4 cm-thick chocolate-brown interval, below which the sediment color changed to olive gray and greenish gray. In the eastern part of the East Siberian Sea, the sediment was mottled black-olive below 10 cm depth. Iron-manganese concretions were found between 2 cm and 10 cm depth at stations 24, 42, and 43, but were also observed at other stations on the shelf that were not part of this study. Generally, few benthic macrofauna was present and bioturbation was weak or absent. Benthic macrofauna, when present, consisted mainly of brittle stars, isopods, few polychaetes, and rare bivalves. All bottom waters were well-oxygenated with concentrations higher than 190 µmol/l, but the shelf bottom-waters in the East Siberian Sea had generally lower concentrations than in the Laptev Sea and bottom waters on the continental slope had lower oxygen concentrations than on the shelf. Concentrations of bottom-water ammonium ranged between 0.2 µmol/L and 1.8 µmol/L. Generally, the slope stations and the shelf stations nearest to the Lena delta had the highest ammonium concentrations, whereas the other shelf stations showed no clear regional trend other than proximity to the Lena delta. Bottom water dissolved inorganic carbon concentrations varied between 2086 µM (Station 53) and 2598 µM (Station 27), and the stable carbon isotope composition of DIC in the waters overlying the cores were between -0.5 ‰ and -6.5 ‰ vs. VPDB.

### 3.2. Dissolved oxygen, manganese, and iron

We show representative profiles of oxygen concentrations in Figure 2 for the Laptev Sea slope station 1, the Laptev Sea shelf stations 23, 30, 45, and the East Siberian Shelf Sea 53 and 63. Oxygen penetration depths varied between 3 mm at Station 58 and more than 60





mm in all slope sediments (Table 2). For the Laptev Sea slope stations 1, 2, 3, and 4, the
maximum depth of oxygen penetration could not be determined, because at penetration
greater than 60 mm the conical sensor needle opened a hole in the sediment through which
oxygen-containing bottom water could potentially have entered the sediment at depth thereby
artificially extending the oxygen penetration depth. In the slope-to-shelf transects the oxygen
penetration depth decreased from >60 mm off-shore to 10 mm at the most inshore station in
the Laptev Sea and the East Siberian Sea. At the two easternmost shelf stations 58 and 63, we
measured the lowest oxygen penetrations depths, 3 mm and 4 mm, respectively. Evidence for
bioturbation and bioirrigation based on multiple microelectrode profile measurements per
core was rare. Only at Station 48 a clear increase in oxygen concentrations below the
sediment surface was observed, indicative of active bioirrigating macrofauna. However, even
at this station, based on investigations in parallel multicore casts, fauna was not abundant. At
all other stations, oxygen concentrations decreased exponentially with depth. Fitting of the
oxygen concentration profiles to the steady advection-diffusion-transport model (Eq. 1)
yielded fluxes that varied between 0.81 and 11.49 mmol m$^{-2}$ d$^{-1}$ (Table 2). These calculated
$O_2$ fluxes compared well with total oxygen uptake rates calculated from whole-core
incubations using 2D optode sensor spots (Table 3). The good fit between the two methods
also supports the notion that bioirrigation and bioturbation effects from meiofauna and
macrofauna were minor.

In the slope sediment at Station 1, concentrations of dissolved manganese and iron

were low throughout the cored depth interval and below 0.2 and 0.5 µM, respectively. The
exception was a small increase for both elements between 4 and 8 cm depth and 14 and 20 cm
depth to concentrations of less than 3 µM, possibly due to slightly more degradable organic
material in these depth intervals (Fig. 2). A similar concentration profile was found for the
other slope station 4 (data not shown), but here concentrations were below 2 µM throughout
for both dissolved iron and manganese and only slightly higher in the topmost cm. On the
shelf, in the Laptev Sea (Station 23 and 30), concentrations of dissolved manganese and iron
were below 0.3 µM and 1.5 µM, respectively, in the top 2 cm and 3 cm at Station 23, before
increasing to maximum concentrations of 69 and 134 µM. At both stations, metal
concentrations decreased again below the concentration maximum indicating that deeper
buried sediment was not a source of the metals and that the dominant source of iron and
manganese was reduction in the topmost 5 cm of sediment. There was a general trend of
increasing manganese concentrations from west to the east. At Station 30 in the Laptev Sea,



the concentration of dissolved manganese was less than 0.3 µM in the topmost cm, but

increased steeply before increasing to maximum concentrations of 189 µM at 9 cm depth.

Similarly, dissolved iron concentrations were below 1 µM to 4 cm depth and then increased to

144 µM. Again, below the maximum, both iron and manganese porewater concentrations

decreased with increasing sediment depth. Even higher iron and manganese concentrations

were found in the East Siberian Sea (Stations 45, 53, 63), where dissolved manganese already

increased from the bottom water to concentrations of 20 µM in the topmost centimeter of

sediment, and iron increased to above 1 µM below 2 cm depth. The steepest manganese

concentration gradient was found at Station 63 in the easternmost East Siberian Sea, where

concentrations were 501 µM in the first cm of sediment with a concentration maximum of

548 µM at 2.5 cm depth and decreasing below this depth to 115 µM at 30 cm depth. Station

63 differed with respect to dissolved iron concentrations from the other stations, because here

dissolved iron showed two small peaks at 3.5 cm and 7 cm, and concentration increased

substantially only below 17 cm depth to concentrations of 189 µM.

### 3.3. $^{35}$S-Sulfate reduction rates and sulfate

Sulfate concentrations showed minor depth gradients at all sampling sites (Fig. 3) and

decreased from starting concentrations between 23.9 mM and 28.1 mM by 0.4 mM to 2.5 mM

from the top to the bottom of the cores. At all stations, turnover of $^{35}$S-tracer was recorded

from the topmost sediment interval to the bottom of the core indicating active bacterial sulfate

reduction (Fig. 3). Depth-integrated rates over the recovered core lengths varied between 0.03

and 1.41 mmol m$^{-2}$ d$^{-1}$ (Table 2). The integrated rates were lowest at Station 1 at 3146 m in

the Laptev Sea and highest at the Station 63 in the easternmost East Siberian Sea. Across the

shelf, depth-integrated rates increased from the west to the east. Example depth profiles of

depth-specific sulfate reduction rates are shown in Fig. 3 for the same six stations as

previously. At Station 1, these rates ranged from 0.03 to 0.38 nmol cm$^{-3}$ d$^{-1}$. At this station,

the variability between replicate cores was large, which is attributed to the fact that many

rates were near the detection limit in our handling procedure. Overall, sulfate reduction was

higher in the top 10 cm of sediment, but showed no pronounced change with depth at this

station. This suggests that the reactivity of the organic material did not change substantially

over the cored depth interval. The second slope station, Station 4, showed a similar rate-depth

profile than Station 1. Depth profiles for the mid-outer shelf stations 23 to 63 all showed



broad sub-surface maxima between 2.5 and 17.5 cm, but the depths of the rate maxima
differed between the different stations (Fig. 3). Peak rates varied between 0.6 nmol cm$^{-3}$ d$^{-1}$ at
Station 30 and 39 nmol cm$^{-3}$ d$^{-1}$ at Station 63. The second highest rate, 7.6 nmol cm$^{-3}$ d$^{-1}$, was
found at the station nearest to the Lena delta, Station 23. At all stations, sulfate reduction rates
decreased from the maxima to rates below 1 nmol cm$^{-3}$ d$^{-1}$ or to below the detection limit at
the bottom of the cores. A particularly sharp decrease in the sulfate reduction rate was
observed between 8 and 9 cm at Station 63, where rates dropped from 8.5 to 0.1 nmol cm$^{-3}$
day$^{-1}$ over 1 cm depth. Since sulfate was abundant throughout the cored intervals, this order-
of-magnitude decrease indicates substantial changes in the reactivity of buried organic matter.
Although no abrupt change in grain size or organic carbon was observed in this core, it is
likely that a historical change in organic sedimentation had taken place during deposition
across this time interval.

**3.4. Porewater dissolved inorganic carbon (DIC), ammonium (NH$_4^+$), and δ$^{13}$C$_{DIC}$**

Porewater concentrations of dissolved inorganic carbon (DIC) and ammonium (NH$_4^+$)
increased with depth at all stations (Fig. 4). The decrease of DIC was between 0.6 mM
(Station 23) and 7.2 mM (Station 50) over the cored sediment depths and ammonium
concentrations increased between 16.8 µM (Station 1) and 549 µM (Station 50). The
steepness of the depth gradients was consistent with the rates of oxygen uptake and bacterial
sulfate reduction for the different stations. The porewater pattern at Station 63 is an exception,
because this station had the highest oxygen uptake and the highest sulfate reduction rates of
all stations, but showed only a modest increase in DIC and NH$_4^+$ concentrations by 1.5 mM
and 57 µM, respectively, over the cored sediment depth. This apparent discrepancy can be
explained by the very low rates of sulfate reduction and sedimentation below 10 cm depth.
Since these deeper layers have not produced large amounts of DIC and NH$_4^+$, only the surface
10 cm contribute significantly to total carbon mineralization and ammonium production.
DIC/NH$_4^+$ ratios of remineralized inorganic carbon and NH$_4^+$ were corrected for the bottom
water DIC and NH$_4^+$ concentrations. For the anoxic parts of the sediment, DIC/NH$_4^+$ ratios
varied between 9.8 for Station 24 and 65 for Station 1, with an overall mean DIC/NH$_4$ ratio of
13.4. The δ$^{13}$C values of DIC consistently decreased with sediment depth indicating the
addition of $^{13}$C-depleted remineralized carbon to DIC. The greatest downcore depletion in $^{13}$C



was observed at Stations 45, 48, and 50, where $\delta^{13}C$ of DIC decreased from -2.0 ‰ near the
sediment surface to -13.9, -16.4, and -18.6 ‰ at the bottom of the cores (Fig. 4).

**4. Discussion**
**4.1. Modelled oxygen, iron, and manganese reduction rates**
Results of the reaction transport modelling of dissolved oxygen, iron, and manganese
concentration profiles for Station 23 are shown in Fig. 5. $O_2$ consumption rates exceeded
sulfate, iron, and manganese reduction rates by a factor of more than 100 (Fig. 5). For the
shelf stations, most of the carbon oxidation therefore takes place in the topmost 5 mm. The
reaction rate profiles for iron and manganese reduction indicate that manganese reduction
dominates in the topmost 2 cm of sediment followed by co-existing iron and sulfate reduction
below (Fig. 5). These observations are consistent with results from the northern Barents Sea
by Vandieken et al. (2006) and Nickel et al. (2008). Optimal fits of the concentration profiles
required a sediment mixing coefficient of 1 x $10^{-4}$ $cm^2$ $sec^{-1}$ in the topmost 2 cm of sediment
at Stations 23 and 53. For the other stations, optimal fits required no sediment mixing by
bioturbation or advective porewater transport by bioirrigation. This result is consistent with
the modelling results of the oxygen microelectrode profiles and the low numbers of
bioturbating macrofauna in the outer shelf sediment. Bacterial sulfate reduction was detected
already at a depth where the sediment was still brown indicating abundant iron
oxyhydroxides. It is therefore likely that the modelled negative iron production rates at the
sediment surface indicate iron oxidation in the mixed upper layer. This pattern was not
observed for manganese, which is consistent with incomplete manganese oxidation at the
sediment surface and loss of dissolved manganese to the bottom water. The latter observation
supports the assessment by Macdonald and Gobeil (2012) that Arctic shelves can export
dissolved manganese to the Arctic interior. Coexistence of net iron reduction and sulfate
reduction at the same depths make it difficult to quantify how much of the iron reduction is
coupled to heterotrophic carbon oxidation and to the re-oxidation of sulfide produced from
bacterial sulfate reduction. Qualitatively, the presence of dissolved iron throughout the
measured porewater profile implies that iron reduction exceeded concomitant sulfate
reduction, iron sulfide precipitation, and reoxidation reactions, which supports the assessment
of net heterotrophic iron reduction. However, previous investigations of the importance of
iron and manganese reduction in Arctic shelf sediments have emphasized the importance of





directly coupled redox processes between iron and manganese (. It is also important to note
that iron and manganese oxyhydroxides can sorb  $Mn^{2+}$ and $Fe^{2+}$ (Canfield et al. 1993). The
concentrations of dissolved $Fe^{2+}$ and $Mn^{2+}$ may therefore underestimate the actual
concentrations of the reduced forms in these sediments.
Depth-integration of the modelled iron and manganese reduction rates and recalculation of the
depth-integrated rates into carbon mineralization equivalents were performed and the
respective carbon oxidation equivalents for these electron acceptors were calculated using an
idealized $(CH_2O)_x$ stoichiometry for organic matter (Vandieken et al., 2006; Nickel et al.,
2008). These rates were then used to calculate the contribution of the different aerobic and
anaerobic electron acceptors to total carbon mineralization for 5 stations on the Laptev and
East Siberian Sea shelf (Table 3). Accepting the simplifications and limitations of porewater-
based rate calculations mentioned above, manganese and iron reduction contributed between
2.3 and 23.7% to the total anaerobic carbon mineralization and between 0.3 and 2.3% to the
total carbon mineralization. Although these numbers may somewhat underestimate the
contribution of metals to carbon mineralization as discussed above, our results suggest that
bacterial sulfate reduction is by far the major anaerobic carbon mineralization pathway. This
conclusion is consistent with the assessment made by Vandieken et al. (2006) and Nickel et al
(2008), who suggested that more ice-free stations in the northern Barents Sea supported
higher of sulfate reduction than the more permanently ice-covered stations reflecting lower
carbon export production.

### 4.2. Marine versus terrestrial organic matter contribution

Terrestrial organic carbon sources to the Laptev and East Siberian shelf and slope are

riverine discharge and coastal erosion of ice core complex (Stein and Macdonald, 2004; Vonk
et al., 2012; Rachold et al., 2004; Fahl and Nöthig, 2007; Semiletov, 1999). Marine organic
carbon is derived from open-water production during the ice-free months, export of ice algae,
and new production in polynyas (Sakshaug et al., 2004; Nitishinsky et al. (2007). Generally,
marine productivity in the Laptev Sea is low and controlled by the nutrient concentrations
derived from Atlantic water, but spring outflow from the Lena River provides an additional
temporary land-derived nutrient source (Pivovarov et al., 1999; Sakshaug et al., 2004;
Nitishinsky et al., 2007; Bourgeois et al., 2017) during late spring ice melt (Raymond et al.,
2007). Terrestrial-derived nutrients can also affect marine productivity either directly by new





production, or indirectly, due to plankton production from remineralized terrestrial DOC/POC (Alling et al., 2012). In the eastern East Siberian and Chukchi Sea, the inflow of nutrient-rich Pacific water supports higher marine primary productivity (e.g., Grebmeier et al., 2006). Ice-rafted transport and bottom boundary layer transport are the two most important modes of particle transport (Wegner et al., 2005; Bauch et al., 2009). Since all sediments sampled in this study were fine-grained siltey clays and clayey silts, coarse-grained woody, ice-rafted material played only a minor role for deposition of organic matter on the outer shelf and slope sediment. The transport direction of inner shelf sediments has been suggested to follow the predominant atmospheric regime, which is thought to be linked to the Arctic Oscillation (AO) (Dimitrenko et al., 2008; Guay et al., 2001; Weingartner et al., 1999). During positive AO southwesterly winds lead to generally eastward transport and repeated inshore transport in the BBL, whereas negative AO favors southerly winds and a predominantly northward transport (Guay et al., 2001; Dmitrenko et al., 2008). Offshore transport of dissolved and particulate organic matter from the Lean delta to the north can occur with the Transpolar Drift, but terrestrial organic material is also transported eastward and obliquely offshore with the Siberian coastal current receiving additional organic material from the Indigirka and Kolyma rivers (Guo et al., 2007; Dudarev et al., 2006). East of 140ºE, the influence of Pacific-derived nutrient-rich water supporting marine production is stronger the further east and offshore the sampling stations are located (Semiletov et al., 2005) (Fig. 1).

Carbon degradation rates in the sediment across the whole Siberian shelf and slope reflect this temporally and spatially diverse distribution of nutrient availability, ice cover, sediment deposition, and current flow regime (Rachold et al., 2004; Dudarev et al., 2006; Semiletov et al., 2005; Sakshaug et al., 2004; Dmitrenko et al., 2005). The proportion of degradable marine-derived organic material at the eastern Stations 50 to 63 on the East Siberian shelf is higher than at the western stations in the Laptev Sea, in line with higher nutrient availability due to the Pacific influence. Ice-free conditions and the opening of water due to northward migration of ice shortly before the sampling likely supported new algal primary production at the shelf stations closest to land leading to enhanced export and deposition on the seafloor. During the time of sampling, only Stations 6 to 27 were ice-free, while Stations 23 and 24 had the longest ice-free condition before sampling. By contrast, Stations 30 to 63 were still covered by ice during sampling. New export of reactive organic material explains why $O_2$ uptake rates were the highest at stations 23 and 24 along the shelf to slope transect from station 1 to station 24 (Boetius and Damm, 1998). The same pattern as for



the $O_2$ uptake rates is also observed for the sulfate reduction rates indicating that reactive
organic matter is also buried below the oxygen penetration depth and mixed layer into the
sulfate-reducing zone. This indicates that a greater portion of reactive organic material is
buried closer to the Lena delta.
Published organic carbon budgets for the Arctic shelves infer an average burial
efficiency of about 1% of exported marine OC (Stein and Macdonald, 2004), while terrestrial
organic carbon, only accounting for about 10% of the organic carbon delivered to the Arctic
Ocean bottom, has been suggested to be preserved with about 90% efficiency (Macdonald et
al., 2015). Recently, Semiletov et al. (2016) compiled a large dataset indicating substantial
aragonite undersaturation of Arctic shelf bottom waters from the Laptev, the East Siberian,
and the Russian part of the Chukchi Sea, which was interpreted due to the remineralization of
terrestrial organic matter. The observation of strongest aragonite undersaturation in the
bottom waters supports a sediment-derived $CO_2$ source or a stagnant bottom boundary layer
(Semiletov et al., 2013). It is therefore possible that oxic carbon mineralization in the topmost
mm of sediment is a major $CO_2$ source for the overlying water.
In order to determine the mineralized proportion of terrestrial and marine organic
matter in the sediment directly, we use the DIC concentration and the carbon isotope
composition of DIC as indicators of remineralized organic matter in the sediment. The
remineralized fraction (F) of $DIC_{total,\ depth\ x}$ at the different sediment depths (x) was defined as
$F = (DIC_{total,\ depth\ x} - DIC_{bottom\ water\ overlying\ core})/DIC_{total,\ depth\ x}$         (3)
and this fraction was plotted against the respective carbon isotope composition of $DIC_{total,\ depth}$
$_x$ (Fig. 6B). The y-intercept of the linear regression was constrained by the carbon isotope
composition of DIC in the bottom water for the respective station so that the slope of the
regression line was the only unknown in this analysis. The gradients for each station yield the
average stable carbon isotope composition of the remineralized organic matter in the sediment
assuming no or very minor isotope fractionation during the oxidation of organic matter. This
calculation assumes that porewater removal of DIC by diagenetic processes such as $CaCO_3$
precipitation was minor and time-invariant, which is supported by the observation that $Ca^{2+}$
and $Mg^{2+}$ porewater concentrations at these shelf and slope stations did not change
significantly with depth (Sun and Brüchert, unpubl. data). Fig. 6 shows exemplary gradients
of the regression for the six stations presented before and Table 3 lists the derived carbon
isotope compositions of remineralized organic matter for all stations. The range of $\delta^{13}C$ of



remineralized DIC varied between -18.8 ‰ ± 1.1 ‰ (Station 53) and -35.8‰ ± 3.0 ‰
(Station 1). The strongly $^{13}$C-depleted isotope composition of -35.8 ‰ for remineralized DIC
at Station 1 suggests the mineralization of strongly $^{13}$C-depleted organic matter and possibly a
strong contribution of terrestrial organic matter to carbon mineralization far offshore, in line
with the very high DIC/NH$_4^+$ ratio of the porewaters at the slope stations. The potential
existence of degradable terrestrial organic matter in slope sediments of 3000 m water depth is
intriguing, since it would imply downslope transport and degradation of terrestrial organic
matter. Northward off-shelf transport of terrestrial organic matter with the Transpolar Drift is
a viable transport mechanism. The contribution of degradable terrestrial organic matter to DIC
in lower slope sediments is also supported by the observation of terrestrially derived
biomarkers in porewater DOC of central Arctic Ocean sediment analyzed by FT-ICRMS
(Rossel et al., 2016) and deep-water sediment trap data in the central Arctic Ocean (Fahl and
Nöthig, 2007), but requires further investigation.
The isotope composition of the remineralized DIC therefore reflects mineralization of
a mixture of organic molecules of different origins – interpreted here as a mixture of
terrestrial and marine-derived organic matter. For the following discussion, given the
uncertainty of the organic matter origin in slope sediment, we exclude data from the slope
stations and restrict the discussion to the use of the following end member compositions for
the shelf stations. For the Laptev Sea shelf, we account for the fact that a fraction of the DIC
used for marine production is derived from remineralized terrestrial DOC and POC in shelf
waters. Alling et al. (2012) report $\delta^{13}C_{DIC}$ values for offshore DIC samples below the
halocline varying between -2 and -4 ‰. We therefore use an isotope endmember for marine
organic matter of -24 ‰ and an isotope composition of -28 ‰ for the terrestrial organic
carbon contribution (Alling et al., 2010; Vonk et al., 2012). For the East Siberian Sea East of
140°E, the heaviest calculated isotope composition of remineralized DIC was -19 ‰ and is
used here as the marine endmember (Station 53). The same carbon isotope composition of -28
‰ as for the Laptev Sea was used as the terrestrial end member. The heavier marine $\delta^{13}$C
value in the East Siberian Sea is supported by the slightly heavier $\delta^{13}C_{DIC}$ values reported for
the offshore East Siberian Sea, which vary between 0 and -2 ‰ (Alling et al., 2012).
The relative contributions of the terrestrial and marine organic carbon were then calculated
with a linear two-endmember isotope model:
$\delta^{13}C_{DIC, \text{ remineralized}} = f_{terr} * \delta^{13}C_{terr \text{ OC}} + f_{mar} * \delta^{13}C_{marine \text{ OC}}$        (4)



where $f_{terr}$ and $f_{mar}$ are the respective mass fractions of terrestrial and marine-derived organic
carbon and $\delta^{13}C_{terrestrial}$ and $\delta^{13}C_{marine}$ reflect the isotope composition of these endmembers.
The calculated mass fractions of the two endmembers are listed in Table 3. Based on this
analysis the porewaters on the Laptev Sea shelf contain a significant proportion of terrestrially
derived organic matter, comprising on average 36 % of the remineralized DIC. This
proportion decreases to average values of 32% in the East Siberian Sea, in line with a greater
marine production in this area due to the inflow of Pacific water (Semiletov et al., 2005,
Dudarev et al., 2006; Naidu et al., 2000).

In order to derive specific degradation rates of the marine and terrestrial carbon

fractions, the endmember mixing-based assessment of the marine and terrestrial organic
carbon contributions to DIC were combined with the $^{35}$S-sulfate reduction rates. Since $^{35}$S-
sulfate reduction rates constrain most of the anaerobic carbon mineralization of sediment
buried below the oxygen penetration depth, our assessment includes, in contrast to earlier
studies, the mineralization rates of terrestrial organic matter beyond the short time period of
oxygen exposure in the topmost mm of sediment. Using sedimentation rates of 0.8 mm $y^{-1}$ for
the outer Laptev Sea (Strobl et al., 1988) and 1.4 mm $y^{-1}$ for the outer East Siberian Sea
(Bröder et al., 2016b), the recovered sediments record a time interval of 250 to 700 years
since burial. Using the mass fractions of terrestrial and marine-derived organic carbon listed
in Table 3, respective mineralization rates of the terrestrial and marine carbon fractions were
calculated from the product of the mass fractions and the depth-integrated $^{35}$S-sulfate
reduction rates (Table 3). This approach is only applicable in combination with depth-
integrated anaerobic carbon mineralization rates, but would be biased if used in combination
with total oxygen uptake rates. The reason for this is that the depth of oxygen penetration
varied only between a millimeter to little more than a centimeter on the shelf, whereas the
corresponding DIC concentrations, even in the topmost centimeter of sediment, are affected
by diffusive exchange along the 30 cm-long concentration profile smoothing out depth-
dependent changes in the source signal of organic matter. It is therefore not possible to assess
the relative fractions of terrestrial and marine organic matter mineralized for discrete depth
intervals. Our combined radiotracer and DIC stable isotope approach suggests that both
marine and terrestrial organic matter are degraded in the buried sediment and that both pools
contribute to degradation products in anoxic buried sediment. This assessment is a significant
modification to earlier studies by Boetius and Damm (1998) and Bourgeois et al. (2017), who



have described organic matter mineralization in Siberian Arctic sediments largely as a

function of oxygen uptake.

Carbon mineralization rates measured along the transect near 130ºE (Stations 1 through

24) reflect the influence of gradual offshore transport of terrestrial organic material (Bröder et

al., 2016a) (Fig. 7A, B). A comparison with the oxygen uptake rates reported by Boetius and

Damm (1998) indicates that all rates measured in 2014 were significantly higher than the rates

measured in 1993 by Boetius and Damm (1998). Although the different rates may reflect a

seasonal effect since Boetius and Damm's data were acquired later in the year than our data,

the increase may also point to higher organic carbon mass accumulation rates compared to 20

years ago, consistent with a decrease in the annual ice cover over the past 20 years in the

Arctic (Arrigo and van Dijken, 2011; Stroeve et al., 2012; Walsh et al., 2017). Whether these

rates reflect higher marine and/or terrestrial accumulation cannot be answered satisfyingly

with this data set.

Fig. 8A compares the oxygen uptake rate of the stations of this study with averaged

oxygen uptake rates from the literature for different shelf, slope, and abyssal plain

environments worldwide (Canfield et al., 2005). The data suggest that there is no significant

difference in the oxygen consumption rates between the Siberian shelf and slope and other

continental margin environments. [35]S-sulfate reduction rates in Sea Siberian slope sediment

are also comparable rates to those in other slope environments (Fig.7B and 8B), but the

sulfate reduction on the shelf are significantly lower, by a factor up to 15. Another difference

apparent from this comparison is the small range in sulfate reduction rates for the outer shelf

and continental slope sediments of the Siberian Arctic (Fig. 8B). This similarity is noteworthy

for several reasons: 1) it suggests that the kinetics of anaerobic carbon degradation in the shelf

and slope sediments reflect similar reactivity of the organic matter. This is surprising since

accumulation rates on the continental slope are significantly slower than on the outer shelf. 2)

The absolute magnitude of the sulfate reduction rates in shelf and slope sediment indicate

significant rates of organic matter mineralization long after burial consistent with the

substantial DIC flux and the strongly [13]C-depleted DIC carbon isotope composition. Overall,

the data that organic matter reactivity substantially changes during burial in shelf sediment,

but that the reactivity of transported organic matter that is exported to deep water across the

shelf does not decrease significantly supporting long-term slow mineralization rates in the

slope environment. Accumulation of the organic material on the slope may therefore be

related to rapid downslope transport of organic material or a rapid offshore transport, e.g., due





to transport with ice or as bottom nepheloid layers cascading from the shelf edge (Ivanov and
Golovin, 2007).

**4.3. Assessment of carbon burial efficiency**

Reported [210]Pb-based sediment accumulation rates in outer shelf Siberian shelf sediment

range between $0.05 \pm 0.02$ g cm[-2] y[-1] in the Laptev Sea (Strobl et al., 1988) and 0.24±0.04 g
cm[-2] y[-1] in the East Siberian Sea (Bröder et al., 2016b). Given surface sediment organic
carbon concentration for this area between 1% and 1.5%, the resulting organic carbon mass
accumulation rates vary between 1.1 mmol m[-2] d[-1] and 1.7 mmol m[-2] d[-1] for the Laptev Sea
(area near Station 23) and 5.5 and 8.2 mmol m[-2] d[-1] in the East Siberian Sea (data for Station
63). We estimated the burial efficiency of terrestrial organic carbon from the ratio of the
depth-integrated sulfate reduction rates relative to the [210]Pb mass accumulation rate of organic
carbon. This treatment assumes that the reported organic carbon mass accumulation rates
largely reflect the refractory component of organic matter. While it is possible that a fraction
of terrestrial and marine organic matter is degraded on shorter time scales than captured by
the [210]Pb method, we assume that the fraction of highly reactive terrestrial organic matter
missed in this treatment is small. The resulting burial efficiency of the terrestrial carbon
fraction is on average $69 \pm 28$ % in the Laptev Sea and $79 \pm 6$ % for the East Siberian Sea.
We also calculated apparent degradation rate constants of organic matter assuming first order
degradation kinetics for the time duration of sediment burial recorded in the sediment cores.
For this assessment, we used the total depth-integrated anaerobic carbon mineralization
determined from the combined manganese, iron, and sulfate reduction rates for the recovered
sediment. The apparent annual degradation rate constant (k) was then calculated from
$$k_{terrestrial} = \frac{\left(-ln\frac{\int_0^{30} OC_{accumulation} - \int_0^{30} OC_{total\ mineralization}}{\int_0^{30} OC_{accumulation}}\right)}{t_{burial}}$$    (5)
where the integrals of OC$_{accumulation}$ and OC$_{mineralization}$ cover a period of 250 years to 700 years
based on the [210]Pb mass accumulation rates. The resulting annual degradation rate constant
($k_{terrestrial}$) ranges between 1 x 10[-4] y[-1] and 5 x 10[-4] y[-1] averaging 1.5 x 10[-4] y[-1] in the Laptev
Sea and between 8 x 10[-5] and 3 x 10[-4] y[-1] averaging 1.2 x 10[-4] y[-1] in the East Siberian Sea.

A comparison of the total oxygen uptake with the C$_{org}$ mass accumulation rates

indicates that the [210]Pb-based C$_{org}$ mass accumulation rates on the shelf are equal or




significantly lower than the oxygen uptake rates, with a discrepancy of up to a factor 10.
Since the derivation of the [210]Pb-based $C_{org}$ mass accumulation rates is based on the same
depth range as our direct [35]S-based degradation rate measurements (30 cm of sediment, Vonk
et al., 2012), $C_{org}$ mass accumulation rates and degradation rate measurements cover the same
time window of sediment burial. Temporal variation in sediment accumulation therefore
cannot explain the discrepancy. In addition, methane seep sediments where upward transport
of methane from deeper sediment layers contributed to oxygen uptake were excluded from
this data set. The best explanation for the discrepancy is therefore that the [210]Pb-mass
accumulation rates underestimate the true mass accumulation rate of highly reactive organic
material and that this material is oxidized at the sediment surface. Based on the measured
oxygen uptake rates this freshly deposited organic material has substantially higher
degradation rates within the top mm of sediment as reflected by the steep $O_2$ gradients. [210]Pb-
based organic carbon accumulation therefore reflects the long-term burial of less reactive
organic material in the top 30 cm of sediment. Since anaerobic degradation processes prevail
below the $O_2$ penetration depth, the measured burial efficiency of the accumulating organic
material is therefore a function of the anaerobic bacterial degradation rather than the aerobic
degradation efficiency. This conclusion has implications regarding the assessment of potential
aerobic degradation of reactive terrestrial organic matter, since degradation of such material
would have gone undetected with [210]Pb-based accumulation rate measurements.

**4.4. The relative importance of iron, manganese and sulfate reduction for carbon**

**degradation**

There was a statistically significant positive correlation between the dissolved oxygen

uptake and anaerobic carbon degradation rates by sulfate reduction with an $r^2$ of 0.72 (P <
0.05). This reflects the coupling of oxygen uptake to the oxidation of reduced inorganic
metabolites (FeS and $H_2S$) produced during the anaerobic metabolism by sulfate reduction
(e.g., Glud, 2008; Jørgensen and Kasten, 2006; Thamdrup, 2000; Berg et al., 2003) (Fig. 9).
The slope of the regression line for the data set is 5.6±1.2 indicating that about 18% of the
oxygen uptake is used for the oxidation of reduced manganese, ammonium, dissolved iron,
and iron sulfides and elemental sulfur. This amount is close to and only slightly lower than the
average 23% estimated for oxygenated coastal and continental shelf sediment (Canfield et al.,
2005), and indicates that a substantial amount of the buried organic matter in Siberian shelf



sediment is oxidized anaerobically. The lower proportion of anaerobic respiration to aerobic
respiration compared to other shelf environments likely reflects the greater proportion of
highly reactive marine-derived organic material in the topmost millimeters of sediment.
Iron hydroxide surfaces have been inferred as important mineral surfaces for the
preservation of organic matter (Lalonde et al., 2012; Salvado et al., 2016). In all cases studied
here, the integrated net DIC production based on the porewater gradient of DIC, and the depth
profiles of iron reduction indicate co-existing heterotrophic and chemical iron reduction and
bacterial sulfate reduction. In addition, the porewater modelling results suggest that
bioturbation is an important sediment mixing process for some shelf stations. Organic matter
sorbed to mineral surfaces with deposition would thus have been subject to repeated
desorption as iron oxyhydroxides were reduced. While this observation does not contradict
the observation that some organic material is buried in association with iron oxyhydroxides,
the repeated cycling of the repeated redox cycling of the oxyhydroxides would prevent the
sorptive preservation of organic compounds.

## 4.5. Regional estimates


We present areal estimates of sediment carbon mineralization by extrapolating the
measured carbon mineralization rates over the outer Laptev Sea and East Siberian Sea shelf.
Such extrapolations of benthic carbon mineralization rates are notoriously difficult given
sediment heterogeneity and insufficient temporal data coverage of benthic carbon
mineralization rates. For this investigation, no near-shore or slope stations were included in
the assessment. The near-shore Siberian shelf environments are under much stronger
influence by coastal erosion and riverine discharge than the outer shelf stations and have
considerable longer open-water conditions than the outer shelf stations investigated here. In
addition, the sedimentation pattern in the near-shore environments is significantly more
diverse, which will affect sedimentation rates, grain size distribution, and carbon contents. For
this reason, we did not extend our extrapolations to the inner shelf environments. Some of
these inner shelf settings likely have much higher benthic carbon mineralization rates and
additional studies are required to constrain these better. Our coverage of the slope stations is
insufficient for meaningful spatial extrapolations. A large data set for this region has been
analyzed by Miller et al. (2016) and the reader is referred to this work.



We estimate the extent of the outer shelf area with depositional conditions comparable

to those investigated here to cover approximately 280,000 km$^2$ of the Laptev Sea. For the East
Siberian Sea, we estimate the respective area of the outer shelf to be 340,000 km$^2$. Due to the
stronger terrestrial influence in the Laptev Sea, we calculated rates separately for the two shelf
seas. The areal coverage with sediment stations was too sparse for statistically significant
interpolations between stations that would give reliable spatial accounts of the gradients in
rates between the stations. Instead, arithmetic averages of sediment mineralization rates and
fluxes were calculated for these regions. Accepting the uncertainties in our assessment and
data density, we estimate that the calculated areal rates could deviate by up to 50%. Table 5
lists the calculated rates based on the average flux calculated per square meter per day for
oxygen uptake, DIC flux, bacterial sulfate, and total anaerobic carbon mineralization. For the
latter three methods, the total flux was calculated for the marine and terrestrial component,
respectively. The same analysis cannot be performed for the oxygen uptake for the reasons
discussed in section 4.1. Since the major part of the oxygen uptake is likely associated with
degradation of a highly reactive marine organic carbon component, the proportions calculated
based on the $\delta^{13}$C composition of DIC would not necessarily apply to the topmost mm of
sediment. It is noteworthy to say that the rates calculated with our data set agree well with the
O$_2$ uptake rates recently published by Bourgeois et al. (2017) for the Laptev Sea. Our
calculations suggest that 5.2 and 10.4 Tg O$_2$ y$^{-1}$, respectively are taken up by the outer shelf
sediment in the Laptev and East Siberian Sea, respectively, totaling 15.9 Tg y$^{-1}$ for the whole
investigated area. Anaerobic carbon mineralization based on DIC, $^{35}$S-SRR and combined
manganese, iron, and sulfate reduction range between 0.62 and 1.28 Tg y$^{-1}$. Of the total
anaerobic carbon mineralization, between 0.25 and 0.48 Tg y$^{-1}$ can be attributed to the
oxidation of terrestrially derived organic material. This rate is five to ten times lower than the
estimated annual water column degradation of particulate terrestrial organic matter in the
Eastern Siberian Arctic shelf system of 2.5±1.6 Tg y$^{-1}$ (Sanchez et al. 2011), and only
between 0.5% and 2% of the annual organic carbon export from land (Stein and Macdonald,
2004; Vonk et al., 2012).

**5. Conclusions**

Directly measured carbon mineralization rates together with stable isotope and

concentration data of East Siberian Arctic shelf and slope porewaters indicate that about one




third of the remineralized organic carbon in porewater DIC is derived from terrestrial organic
matter. This conclusion confirms and extends previous observations that terrestrial organic
carbon buried in Siberian shelf and slope sediment is not conservative (Semiletov et al., 2013;
Karlsson et al., 2015; Bröder et al., 2016b). While mineralization of terrestrial organic
material has been described for the water column and resuspended surface sediment, our data
indicate that mineralization also proceeds long after burial in sediment. The estimated
apparent carbon degradation rate constants of transformed terrestrial organic matter on the
outer shelf are slow ($< 3 \times 10^{-4}$ $y^{-1}$) and the overall terrestrial carbon burial efficiency is
relatively high ($> 69$ %), but lower than previously reported based on millennial-scale carbon
burial rates (90 %, Stein and Macdonald, 2004). The low degradation rates are most apparent
in the low bacterial sulfate reduction rates, which is the major anaerobic electron acceptor in
these sediments. High porewater concentrations of dissolved iron and manganese testify to
significant iron and manganese reduction that contributed up to 23.7% to the anaerobic carbon
mineralization in these sediments, but only 2.3% to the total carbon mineralization. The
pervasive presence of dissolved iron at all but one of the sediment stations in the eastern East
Siberian Sea indicates the presence of abundant iron oxyhydroxides, which possibly play a
role in the preservation of organic matter (Salvado et al., 2016; Lalonde et al., 2012).
The regional differences in the isotope composition of remineralized DIC and
DIC/$NH_4^+$ ratios are in accordance with a greater proportion of terrestrially-derived organic
matter in the Laptev Sea, but the absolute mineralization rates of the terrestrially derived
organic matter were similar in both regions. This observation is important in light of a
potential priming effect by marine-derived organic matter (Bianchi, 2011). Our data do not
indicate that the larger marine fraction in the outer East Siberian Sea had a greater priming
effect on terrestrial carbon than in the outer Laptev Sea, but an overall priming effect may
deduced from the dual contribution of terrestrial and marine-derived organic matter to DIC.
Area-integrated rates of carbon mineralization in the outer shelf sediments (0.29 to 0.48 Tg $y^{-1}$
) represent about 0.5 % to 8 % of the annual terrestrial organic matter load to the Laptev and
East Siberian Sea ranging from 6 Tg $y^{-1}$ (Stein and Macdonald, 2004) to 22±8 to 44 Tg $y^{-1}$
(Vonk et al., 2012). There are large uncertainties associated with these estimates, given that
our calculations do not account for carbon mineralization of resuspended terrestrial organic
material and likely higher rates of mineralization in the inner shelf sediments. Nevertheless,
these data indicate that the contribution of the benthic DIC flux to the total $CO_2$ production in
the outer Eastern Siberian Sea and Laptev Sea is small. This conclusion, however, does not



necessarily extend to the inner parts of the Laptev Sea and the western parts of the East
Siberian Sea, where $CO_2$ supersaturation has been reported by Semiletov et al. (2012) and
Pipko et al. (2011). Anderson et al (2009) estimated a DIC excess of 10 Tg C by evaluating
data from the Laptev and East Siberian Seas collected in the summer of 2008 and suggested
that this excess was caused mainly by terrestrial organic matter decomposition. Their estimate
can be compared to our sediment oxygen uptake for the outer Laptev and East Siberian Sea
shelf of almost 16 Tg $y^{-1}$, which would demand that 62.5 % of the oxygen uptake was due to
terrestrial organic matter mineralization. However, the reported annual production of marine
organic matter for the total Laptev and East Siberian Sea is about 46 Tg $y^{-1}$ (Stein and
Macdonald, 2004). Even if only half of this amount is produced in the outer shelf region and
only another half of that amount was deposited, there would still be more than 10 Tg $y^{-1}$ of
reactive marine organic matter at the sediment surface. Our data would therefore suggest that
at least in the more productive East Siberian Sea the pronounced aragonite undersaturation
reported for bottom waters in the East Siberian Sea is due to aerobic mineralization of a
significant amount of marine organic matter, which extends the assessment for the western
Chukchi Sea and the central Arctic Ocean by Qi et al. (2017). It is apparent that these
sediments play a major role in the recycling of marine organic carbon on the Arctic shelf.
Future changes in marine production on the Siberian shelf under longer ice-free conditions
(Arrigo and van Dijken, 2011) will likely change the relative proportions of degrading marine
and terrestrial organic matter further so that this particular shelf system may in the future
more strongly resemble that of other ice-free shelf-slope environments.

## 6. Acknowledgements

Funding for this investigation came from the K&A Wallenberg foundation, the Swedish
Polarsekretariat, and the Bolin Centre for Climate Research at Stockholm University. Igor
Semiletov acknowledge support from  the Russian Government (No.
14.Z50.31.0012/03.19.2014). We would like to thank the members of the SWERUS-C3
consortium, the shipcrew on icebreaker Oden, and Heike Siegmund, Lina Hansson, Barkas
Charalampos, and Dimitra Panagiotopoulou for help with the laboratory work. We dedicate
this publication to our friend and colleague Vladimir Samarkin, who unfortunately passed
away before publication of this work. This manuscript benefitted from discussions with



Patrick Crill, Örjan Gustafsson, Christoph Humborg, Julia Steinbach, Clint Miller, Marc
Geibel, Emma Karlsson, Brett Thornton, Jorien Vonk, Leif Anderson, and Magnus Mörth.
**List of Tables**

Table 1. Physical and chemical characteristics of sediment and bottom water at the sampled
stations

Table 2. Summary of bottom water concentrations, carbon isotope composition of bottom
water DIC and remineralized DIC, DIC/$NH_4^+$ ratios, $O_2$ uptake, and integrated $^{35}$S-sulfate
reduction rates

Table 3. Calculated carbon isotope composition of remineralized DIC and mass fractions of
the marine and terrestrial end member and corresponding terrestrial carbon degradation rates
based on $^{35}$S-SRR and DIC

Table 4. Anaerobic rates of carbon mineralization by manganese, iron, and sulfate reduction

Table 5. Regional estimates of sediment carbon mineralization in the outer Laptev and East
Siberian shelf sea



**List of figures**
Figure 1: General map of the Laptev and East Siberian Sea with sediment stations and major
current features

Figure 2: Depth profiles of dissolved $O_2$ measured with $O_2$ microelectrode sensors for Stations
1, 23, 30, 45, 58, and 63 and profiles of porewater concentrations of dissolved iron and
manganese.

Figure 3: $^{35}$S-SRR rates and corresponding porewater sulfate concentrations for Stations 1, 23,
30, 45, 58, and 63.

Figure 4: Depth profiles of dissolved inorganic carbon (DIC), $\delta^{13}C_{DIC}$, and dissolved
ammonium ($NH_4^+$) for Stations 1, 23, 30, 45, 58, and 63.

Fig. 5.  Comparison of reaction rates of oxygen, manganese, iron, and sulfate reduction at
Station 23. Note the different depth scale for the $O_2$ consumption rate. The dashed line marks
the oxygen penetration depth.

Fig. 5 A, B. Map of field area and sampling stations showing oxygen uptake rates in panel A
and depth-integrated sulfate reduction rates in panel B. For comparison, oxygen uptake rates
reported in Boetius and Damm (1998) using the same color coding are shown as triangles for
comparison.

Fig. 6. A: Crossplot of dissolved $NH_4^+$ and porewater DIC* after correction for bottom water
DIC concentrations. The slopes of the regression lines for the individual stations are shown in
Table 2. B: Crossplot of the fraction of remineralized DIC calculated from a two endmember



mixing model versus $\delta^{13}C_{DIC}$. The slope and y-intercept of the regression for each station are
shown in Table 3.

Fig. 7 A, B. Map of field area and sampling stations showing oxygen uptake rates in panel A
and depth-integrated sulfate reduction rates in panel B.

Fig. 8A. Water depth variation of sediment oxygen uptake. 8B: Water depth variation of
integrated $^{35}$S-sulfate reduction rates (0-30 cm sediment depth). For reference average rates of
abyssal plain, continental rise, slope, and shelf sediments, deposition and non-depositional,
are shown for reference.

Fig. 9. Crossplot of diffusive oxygen uptake  and integrated sulfate reduction rates. The black
line is the linear regression and yielded a y-intercept of 2.1 mmol m$^{-2}$ d$^{-1}$ and a slope of 5.55.
Blue and red lines show the 95% and 99% confidence interval.




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









**Table 1. Physical and chemical bottom characteristics**

| Station | Latitude °N | Longitude °E | Date Month/Day/Year | Water depth m | Ice cover % | Bottom water salinity ‰ | Bottom water temperature °C | Bottom water O₂ concentration µmol/L | Bottom water NH₄⁺ concentration µmol/L | Bottom water DIC above sediment µmol/L | δ¹³C DIC bottom water ‰ vs. VPDB | Sediment description |
|---|---|---|---|---|---|---|---|---|---|---|---|---|
| 1 | 78.942 | 125.243 | 7/15/2014 | 3146 | 50 - 75 | 34.9 | -0.9 | 271.9 | 1.65 | 2151.5 | -0.5 | clay, chocolate brown |
| 2 | 78.581 | 125.607 | 7/16/2014 | 2900 | 25 - 50 | 34.9 | -0.9 | 275.0 |  | n.a. | n.a. | clay, chocolate-brown |
| 3 | 78.238 | 126.150 | 7/16/2014 | 2601 | < 25 | 34.9 | -0.9 | 280.0 |  | n.a. | n.a. | clay, chocolate-brown |
| 4 | 77.855 | 126.664 | 7/16/2014 | 2106 | < 25 | 34.9 | -0.8 | 289.4 | 1.81 | 2164.5 | -1.6 | clay, chocolate-brown |
| 6 | 77.142 | 127.378 | 7/17/2014 | 89 | 0.0 | 34.6 | -1.8 | 327.0 | 1.30 | 2213.0 | -2.2 | clay, top 3 cm brown, then gray, fauna on top of sediment |
| 23 | 76.171 | 129.333 | 7/22/2014 | 56 | 0.0 | 34.2 | -1.8 | 303.2 | 1.34 | 2246.3 | -3.2 | silty clay, top 4 cm brown, then gray, brittle stars |
| 24 | 75.599 | 129.558 | 7/24/2014 | 46 | 0.0 | 34.0 | -1.7 | 283.8 | 0.89 | 2244.1 | -2.0 | silty clay, top 4 cm brown, then gray |
| 27 | 76.943 | 132.229 | 7/23/2014 | 44 | 0.0 | 34.2 | -1.8 | 332.3 | 0.94 | 2595.0 | -6.5 | silty clay, top 2 cm brown, then gray, fluffy surface layer, brittle stars |
| 30 | 78.181 | 138.354 | 7/24/2014 | 69 | 0.0 | 34.1 | -1.6 | 334.8 | 0.79 | 2178.4 | -3.7 | silty clay, top 4 cm brown, then gray |
| 31 | 79.396 | 135.497 | 7/25/2014 | 3056 | 0.0 | 34.9 | -0.9 | 270.9 | 0.74 | 2161.7 | n.a. | clay, chocolate brown |
| 35 | 78.600 | 137.061 | 7/26/2014 | 541 | 0.0 | 34.9 | 0.4 | 288.1 | 0.43 | 2183.7 | n.a. | clay top 15cm brown, fluffy, inhomogeneous, surface-dwelling fauna |
| 37 | 78.521 | 137.170 | 7/26/2014 | 205 | 0.0 | 34.7 | 0.0 | 295.4 | 0.89 | 2171.1 | n.a. | clay, top 5cm brown, then gray |
| 40 | 77.670 | 144.668 | 7/27/2014 | 45 | 0.0 | 31.5 | -1.3 | 190.3 | 0.53 | 2213.7 | -1.6 | silty clay, top 3cm brown, then gray, brittle stars |
| 43 | 76.780 | 147.791 | 7/28/2014 | 42 | 25-50 | 30.1 | -1.2 | 256.4 | 0.61 | 2086.7 | n.a. | silty clay to clayey silt, top 2cm brown, then gray, some small durface-dwelling animals |
| 45 | 76.416 | 148.115 | 7/29/2014 | 40 | < 50 | 29.1 | -1.3 | 319.9 | 0.57 | 2576.0 | -2.1 | silty clay to clayey silt, 2cm brown, then gray-black, rather stiff |
| 48 | 76.615 | 153.345 | 7/30/2014 | 49 | > 75 | 30.6 | -1.6 | 315.9 | 0.50 | 2075.1 | -2.2 | silty clay to clayey silt, top 3cm brown, then grayblack |
| 50 | 75.764 | 158.529 | 8/1/2014 | 44 | > 75 | 31.1 | -1.4 | 311.0 | 0.51 | 2068.7 | -2.1 | silty clay to clayet silt, top 2cm brown, then grayblack |
| 53 | 74.957 | 161.088 | 8/2/2014 | 47 | > 75 | 31.0 | -1.6 | 253.3 | 0.16 | 2086.1 | -2.5 | silty clay to clayey silt, top 3 cm brown, then 3 cm gray, then grayblack |
| 58 | 74.440 | 166.050 | 8/4/2014 | 54 | > 75 | 31.4 | -1.7 | 254.3 | 0.65 | 2154.9 | -1.5 | silty clay to clayey silt, slightly resuspended, top 2 cm brown, then gray, soft |
| 63 | 74.685 | 172.361 | 8/7/2014 | 67 | > 75 | 32.4 | -1.4 | 186.0 | 0.61 | 2240.8 | -2.2 | silty clay to clayey silt, top 1cm brown, then gray |





**Table 2.** Summary of bottom water concentrations, carbon isotope composition of bottom water DIC and remineralized DIC, DIC/NH$_4^+$ ratios, O$_2$ uptake, and integrated $^{35}$S-sulfate reduction rates

| Station | Bottom water NH$_4^+$ above sediment | Bottom water DIC above sediment | δ$^{13}$C DIC bottom water | mean O$_2$ penetration depth | mean O depth | O$_2$ uptake (modelled) | $^{35}$S-SRR (0-30 cm) duplicates | DIC flux (modelled) | Average porewater DIC/NH$_4^+$ |
|---|---|---|---|---|---|---|---|---|---|
|  | µmol/L | µmol/L | ‰ vs. VPDB | mm | µmol/L | mmol m$^{-2}$ d$^{-1}$ | mmol m$^{-2}$ d$^{-1}$ | mmol m$^{-2}$ d$^{-1}$ |  |
| **1** | 1.6 | 2151.5 | -0.5 | > 60 | 217 | 1.48 ± 0.08 | 0.05 / 0.21 | -0.11 |  |
| **2** | n.a. | n.a. | n.a. | > 60 | 213 | 1.32 ± 0.05 |  |  |  |
| **3** | n.a. | n.a. | n.a. | > 60 | 194 | 0.81 ± 0.06 |  |  |  |
| **4** | 1.8 | 2164.5 | -1.6 | > 60 | 89 | 1.32 ± 0.05 | 0.17 / 0.17 | -0.15 |  |
| **6** | 1.3 | 2213.0 | -2.2 | 36 | 0 | 2.61 ± 0.01 | 0.03 / 0.05 | -0.08 |  |
| **23** | 1.3 | 2246.3 | -3.2 | 13 | 0 | 10.00 ± 0.09 | 0.56 | -0.12 | 13 |
| **24** | 0.9 | 2244.1 | -2.0 | 10 | 0 | 7.95 ± 0.14 |  | -0.22 | 10 |
| **27** | 0.9 | 2595.0 | -6.5 | 16 | 0 | 3.75 ± 0.08 | 0.37 / 0.20 | -0.27 | 12 |
| **30** | 0.8 | 2178.4 | -3.7 | 16 | 0 | 2.61 ± 0.11 | 0.06 / 0.03 | -0.12 | 15 |
| **31** | 0.7 | 2161.7 | n.a. | > 60 | 194 | 1.78 ± 0.07 |  |  |  |
| **35** | 0.4 | 2183.7 | n.a. | > 60 | 30 | 2.43 ± 0.32 |  |  |  |
| **37** | 0.9 | 2171.1 | n.a. | 44 | 0 | 2.51 ± 0.10 |  |  |  |
| **Average Laptev Sea shelf** |  |  |  |  |  | **4.20** | **0.19** | **0.16** | **12** |
| **40** | 0.5 | 2213.7 | -1.6 | 12 | 0 | 4.62 ± 0.08 | 0.33 / 0.24 | -0.19 | 16 |
| **43** | 0.6 | 2086.7 | n.a. | 13 | 0 | 4.7 ± 0.10 |  |  |  |
| **45** | 0.6 | 2576.0 | -2.1 | 10 | 0 | 4.02 ± 0.10 | 0.23 / 0.19 | -0.37 | 13 |
| **48** | 0.5 | 2075.1 | -2.2 | 5 | 0 | 9.14 ± 0.22 | 0.68 / 0.53 | -0.71 | 10 |
| **50** | 0.5 | 2068.7 | -2.1 | 9 | 0 | 8.65 ± 0.43 | 1.32 / 1.40 | -1.01 | 12 |
| **53** | 0.2 | 2086.1 | -2.5 | 10 | 0 | 4.53 ± 0.08 | 0.10 / 0.17 | -0.20 | 14 |
| **58** | 0.7 | 2154.9 | -1.5 | 3 | 0 | 11.49 ± 0.52 | 1.01 | -1.27 | 24 |
| **63** | 0.6 | 2240.8 | -2.2 | 4 | 0 | 10.72 ± 0.15 | 1.41 | -1.35 | 12 |
| **Average East Siberian Sea shelf** |  |  |  |  |  |  | **1.41** | **0.73** | **14** |





**Table 3. Calculated carbon isotope composition of reminaralized DIC and mass fractions of marine and terrestrial end member and corresponding terrestrial carbon degradation rates based on $^{35}$S-SRR and DIC flux**

| Station | Average $\delta^{13}C_{DIC}$ remineralized | Marine end member | Terrestrial end member | $^{35}$S-SRR-based terrestrial degradation rate | DIC-based terrestrial degradation rate |
|---|---|---|---|---|---|
| | ‰ vs. VPDB | Mass fraction | | mmol m$^{-2}$ d$^{-1}$ | mmol m$^{-2}$ d$^{-1}$ |
| 1 | -32.5 | 0.0 | 1.0 | 0.13 | 0.11 |
| 4 | -24.7 | 0.73 | 0.27 | 0.05 | 0.04 |
| 6 | -25.1 | 0.65 | 0.35 | 0.01 | 0.03 |
| 23 | -24.5 | 0.78 | 0.22 | 0.12 | 0.03 |
| 24 | -24.7 | 0.73 | 0.27 | | 0.06 |
| 27 | -25.4 | 0.58 | 0.42 | 0.12 | 0.11 |
| 30 | -28.5 | 0.00 | 1.00 | 0.05 | 0.13 |
| **Average Laptev Sea shelf** | **-25.6** | **0.53** | **0.47** | **0.08** | **0.07** |
| 40 | -21.4 | 0.72 | 0.28 | 0.08 | 0.05 |
| 45 | -22.2 | 0.63 | 0.37 | 0.08 | 0.14 |
| 48 | -23.0 | 0.54 | 0.46 | 0.28 | 0.32 |
| 50 | -24.0 | 0.43 | 0.57 | 0.77 | 0.57 |
| 53 | -18.8 | 1.00 | 0.00 | 0.00 | 0.00 |
| 58 | -22.6 | 0.59 | 0.41 | 0.42 | 0.53 |
| 63 | -20.3 | 0.84 | 0.16 | 0.25 | 0.22 |
| **Average East Siberian Sea shelf** | **-21.8** | **0.68** | **0.32** | **0.27** | **0.26** |







**Table 4 Anaerobic rates of carbon mineralization by manganese, iron, and sulfate reduction**

| | Net Fe$^{2+}$ production | Net Mn$^{2+}$ production | C-equivalent Fe + Mn reduction | $^{35}$S-Sulfate reduction | C-equivalents total anaerobic mineralization | Oxygen uptake | % Fe + Mn reduction of total anaerobic mineralization | Percentage anaerobic C mineralization of total | Percentage Fe and Mn mineralization of total |
|---|---|---|---|---|---|---|---|---|---|
| | mmol m$^{-1}$ d$^{-1}$ | | | | | | | % | |
| Station 23 | 0.05 | 0.03 | 0.03 | 0.56 | 1.1 | 10.0 | 2.3 | 11.5 | 0.3 |
| Station 30 | 0.02 | 0.04 | 0.03 | 0.05 | 0.1 | 2.6 | 21.9 | 4.4 | 1.0 |
| Station 45 | 0.14 | 0.12 | 0.09 | 0.21 | 0.5 | 4.0 | 18.3 | 12.8 | 2.3 |
| Station 53 | 0.15 | 0.09 | 0.08 | 0.14 | 0.4 | 4.5 | 23.7 | 7.8 | 1.8 |
| Station 63 | - | 0.50 | 0.25 | 1.41 | 3.1 | 10.7 | 8.1 | 26.0 | 2.3 |




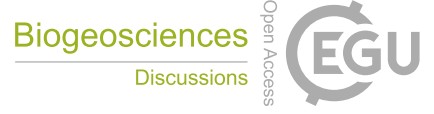

**Table 5. Regional estimates of sediment carbon mineralization in the outer Laptev and East Siberian shelf sea**

| | | | Dissolved O$_2$ uptake | Upward DIC flux (anaerobic) | Terrestrial OC-derived DIC flux (anaerobic) | Marine OC-derived DIC flux (anaerobic) | Depth-integrated $^{35}$S-SRR (C equivalent) |
|---|---|---|---|---|---|---|---|
| Outer Laptev Sea | mmol m$^{-2}$ d$^{-1}$ | Average | 4.2 | 0.16 | 0.07 | 0.09 | 0.09 |
| Outer East Siberian Sea | mmol m$^{-2}$ d$^{-1}$ | Average | 7.2 | 0.73 | 0.26 | 0.47 | 0.34 |
| Outer Laptev Sea | Tg C y$^{-1}$ | 280,000 km$^2$ | 5.2 | 0.20 | 0.09 | 0.11 | 0.11 |
| Outer East Siberian Sea | Tg C y$^{-1}$ | 340,000 km$^2$ | 10.8 | 1.09 | 0.39 | 0.70 | 0.50 |
| Total outer shelf area | Tg C y$^{-1}$ | 620,000 km$^2$ | 15.9 | 1.28 | 0.48 | 0.81 | 0.62 |

| | | | $^{35}$S-SRR-based terrestrial C degradation | $^{35}$S-SRR-based marine C degradation | Total TEAP-based anaerobic OC degradation rate | Total TEAP-based anaerobic terrestrial OC degradation rate | Total TEAP-based anaerobic marine OC degradation rate |
|---|---|---|---|---|---|---|---|
| Outer Laptev Sea | mmol m$^{-2}$ d$^{-1}$ | Average | 0.04 | 0.05 | 0.15 | 0.05 | 0.10 |
| Outer East Siberian Sea | mmol m$^{-2}$ d$^{-1}$ | Average | 0.13 | 0.21 | 0.42 | 0.16 | 0.26 |
| Outer Laptev Sea | Tg C y$^{-1}$ | 280,000 km$^2$ | 0.05 | 0.07 | 0.18 | 0.06 | 0.12 |
| Outer East Siberian Sea | Tg C y$^{-1}$ | 340,000 km$^2$ | 0.20 | 0.31 | 0.62 | 0.23 | 0.39 |
| Total outer shelf area | Tg C y$^{-1}$ | 620,000 km$^2$ | 0.25 | 0.37 | 0.80 | 0.29 | 0.51 |






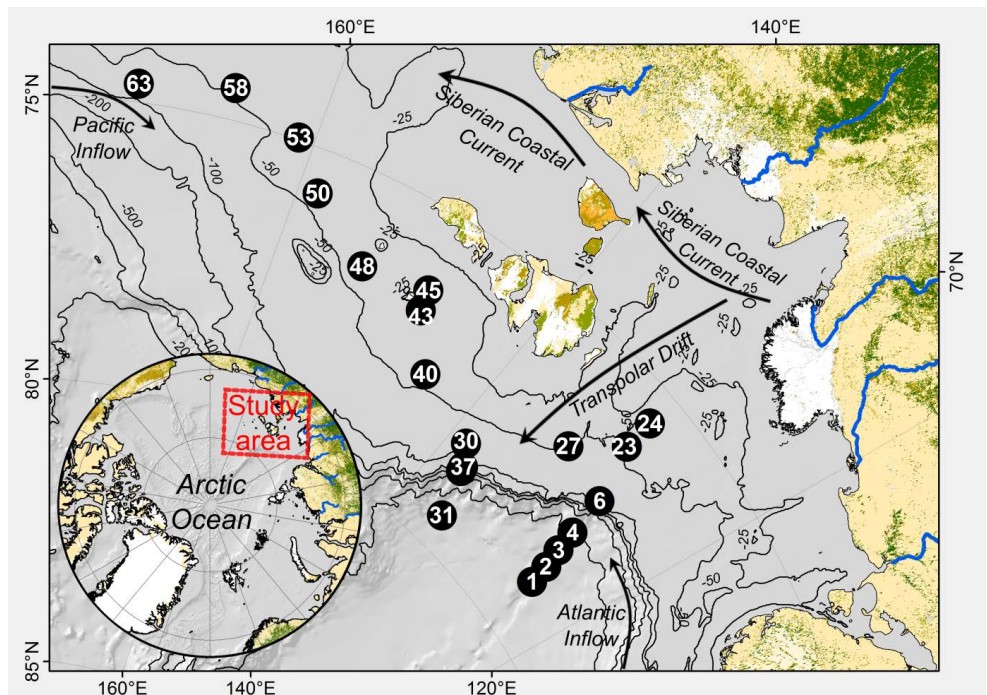


Fig. 1. Map of the Eastern Siberian Sea and slope and station locations.




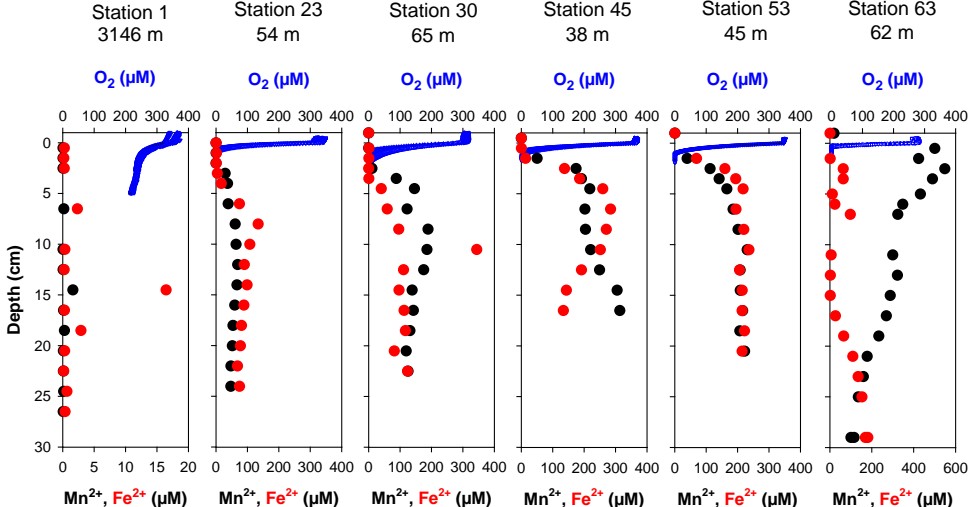


Fig. 2. Depth profiles of dissolved $O_2$, $Fe^{2+}$, and $Mn^{2+}$ at Stations 1, 23, 30, 45, 53, and 63. For
microelectrode profiles, 4 replicates are shown for each station. Depth resolution of measurement
for $O_2$ was 100 μm.




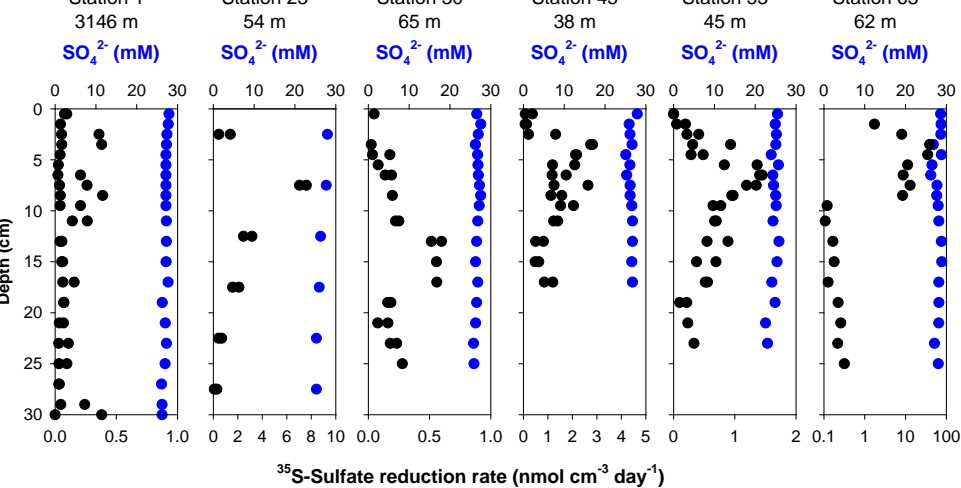


Fig. 3. Depth of profiles of $^{35}S$-sulfate reduction rates and porewater concentration of dissolved
sulfate for Stations 1, 23, 30, 45, 53, and 63. A replicate incubation was conducted for each depth
except for Station 63.






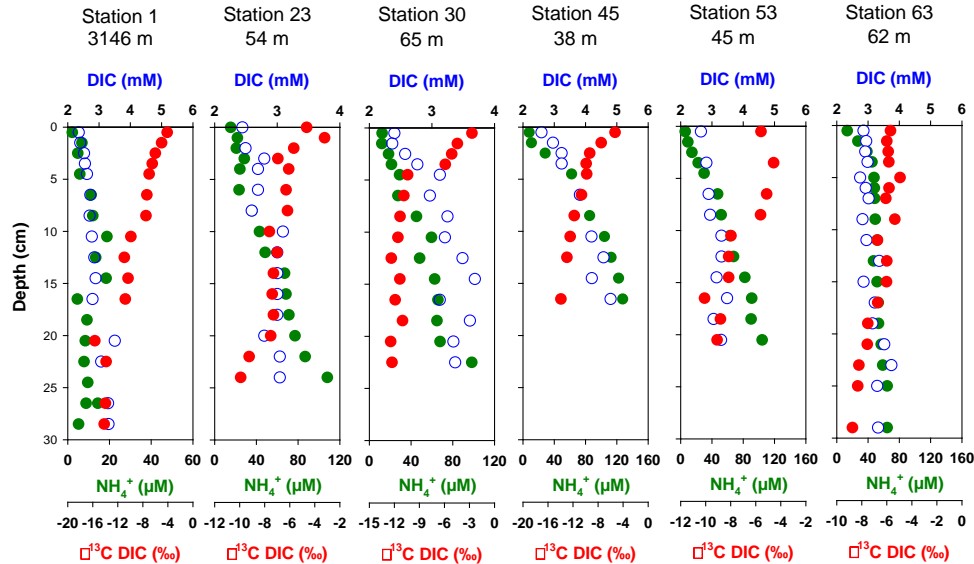

Fig. 4. Depth profiles of porewater dissovled inorganic carbon (DIC), $\delta^{13}$C DIC and porewater $NH_4^+$ at stations 1, 23, 30, 45, 53, and 63.

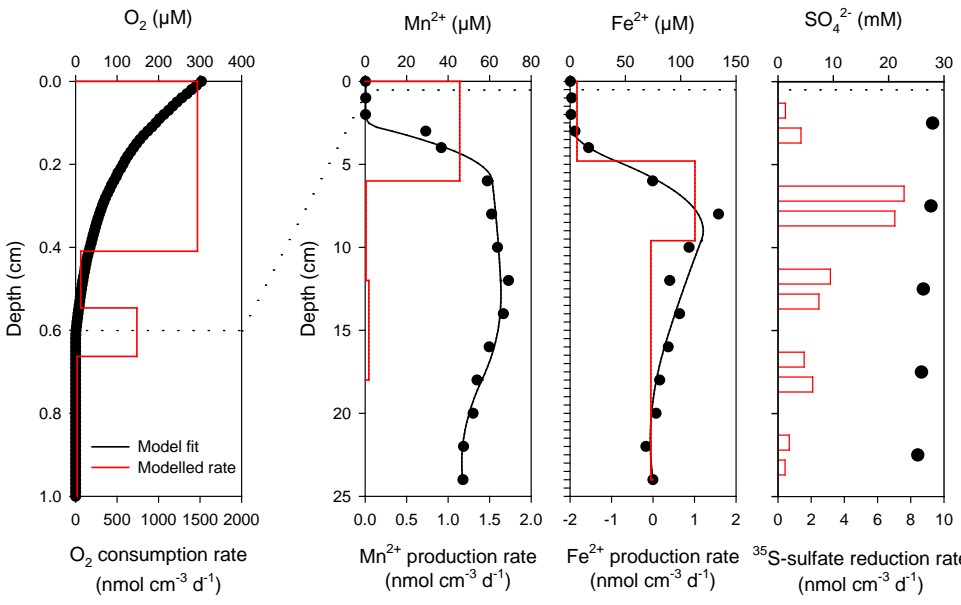

Fig. 5. Comparison of reaction rates of oxygen, manganese, iron, and sulfate reduction at Station 23. Note the different depth scale for the $O_2$ consumption rate. The dashed line marks the oxygen penetration depth.




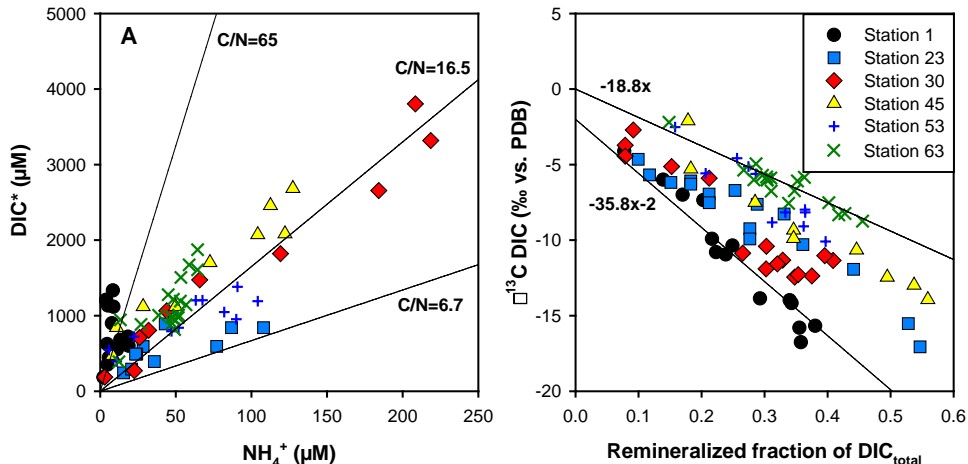


Fig. 6. A: Crossplot of dissolved $NH_4^+$ and porewater DIC* after correction for bottom water DIC
concentrations. The slopes of the regression lines for the individual stations are shown in Table 2. B:
Crossplot of the fraction of remineralized DIC calculated from a 2-endmember mixing model versus
$\delta^{13}$C DIC. The slope and y-intercept of the regression for each station are shown in Table 3.



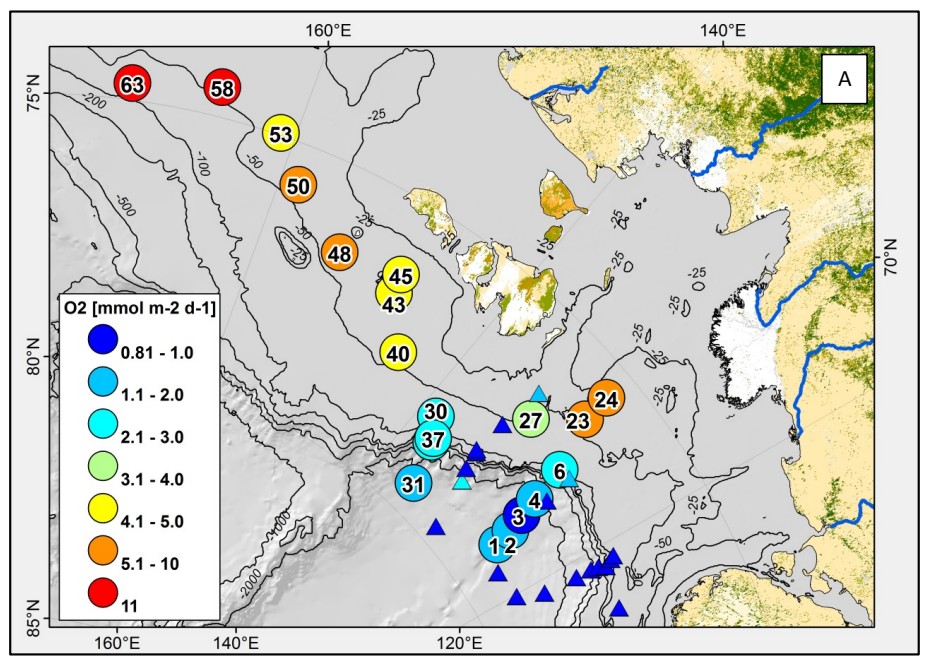


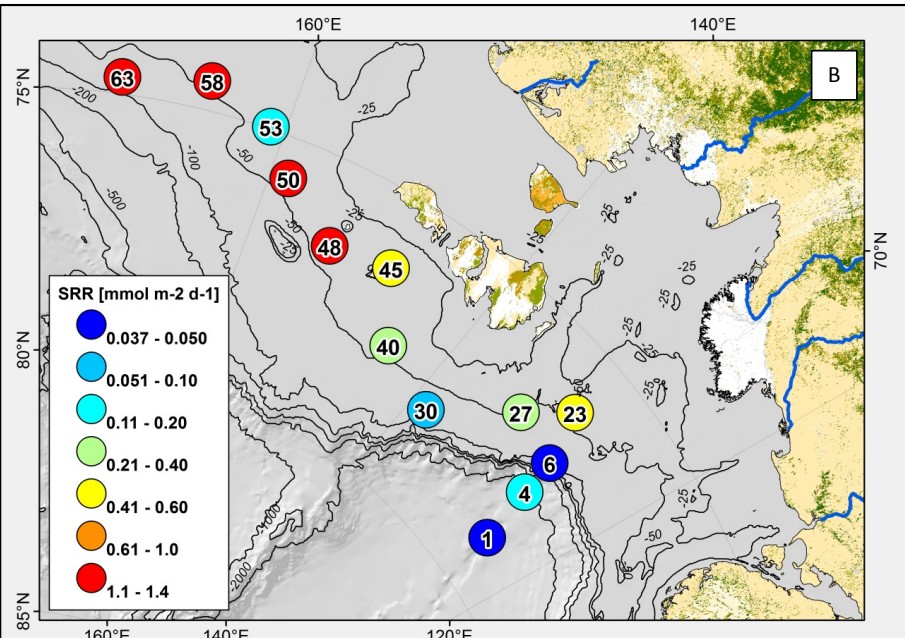


Fig. 7 A, B. Map of field area and sampling stations showing oxygen uptake rates in panel A and
depth-integrated sulfate reduction rates in panel B.







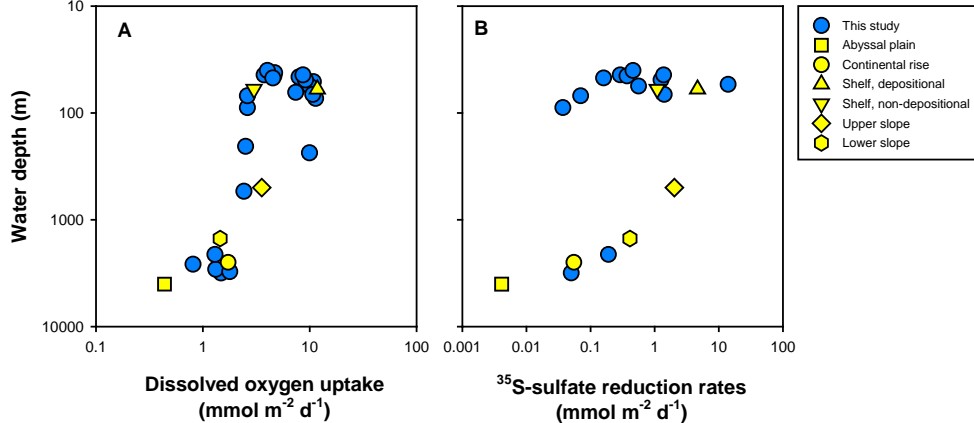


Fig. 8A. Water depth variation of sediment oxygen uptake. 8B: Water depth variation of integrated
[35]S-sulfate reduction rates (0-30 cm sediment depth). For reference average rates of abyssal plain,
continental rise, slope, and shelf sediments, deposition and non-depositional, are shown for
reference.



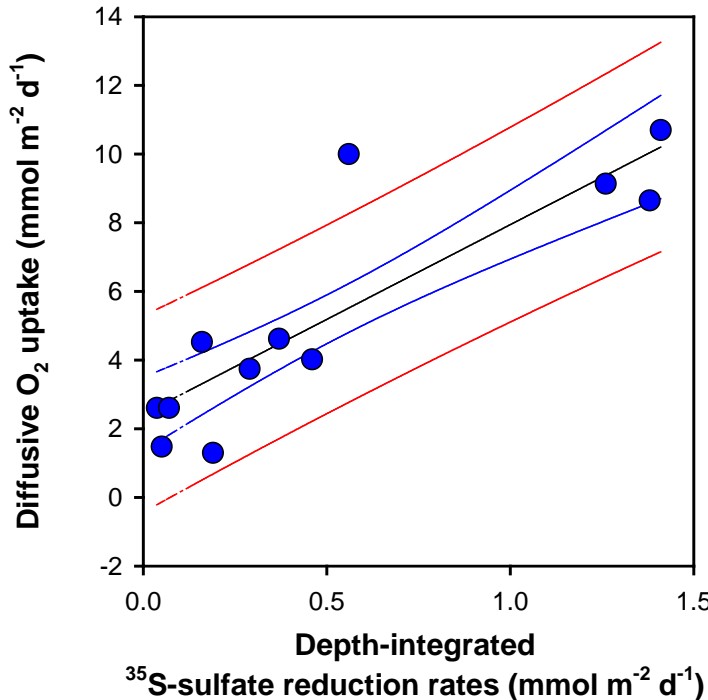


Fig. 9. Crossplot of diffusive oxygen uptake and integrated sulfate reduction rates. The black line is
the linear regression and yielded a y-intercept of 2.1 mmol m$^{-2}$ d$^{-1}$ and a slope of 5.55. Blue and red
lines show the 95% and 99% confidence interval.




