# Peer review of "Carbon mineralization in Laptev and East Siberian Sea shelf and slope sediment"

_Biogeosciences, 2017_

## Referee Comment (RC1) · Anonymous Referee #1 · 7 May 2017

General comments In the manuscript bg-2017-119 presents some important new data on early diagenetic properties and carbon mineralization in the Laptev and East Siberian Sea shelf and slope sediment. In my opinion, it should eventually be published but requires at least moderate revision. Generally, data set is interesting and solid, and conclusions are believable, but the discussion part is a bit unfocussed and manuscript needs to be more concise. In addition, I would suggest that the effect of physical activities such as resuspention and redeposition on pore-water solute concentrations in some shallow sites should be discussed in this manuscript.

Specific comments

(1) line 207 The boundary conditions of the reaction-transport model should be showed in 2.5.

(2) Line 210 Generally, the diffusion process includes molecular diffusion and bioturbation, and is not related to bioirrigation. If the study area has evident bioirrigation process, the bioirrigation term a(C0-C) should be added to the model (a is the irrigation coefficient).

(3) Table 2 showed that DIC fluxes are apparently lower than oxygen uptake rates , but as far as I know, oxygen uptake rates were similar or lower than DIC fluxes in many estuarine and shelf regions. Are there any important processes for removing the pore-water DIC in your study regions? Please explain more about the differences between DIC fluxes and oxygen uptake rates.

(4) Line 371 Pore-water DIC/NH4+ ratios should be corrected by diffusion and adsorption process or at least the authors should demonstrate that these processes can be neglected.

(5) Line 639 In general, the net Corg settling rates equal to sum of 210Pb-based Corg mass accumulation rates and oxygen uptake rates if the sedimentary Corg changed little with depth, and thus 210Pb-based Corg mass accumulation rates can be lower or higher than oxygen uptake rates.

(6) Line 672 Some shelf stations which were influenced by bioturbation should be written, and these discussions about mixing process look like contrasting with the results of optimal fits of the concentration profiles (line 388). Please explain more about the mixing process.

(7) Line747 It is difficult to build relationship with priming effect based on existing data in this paper unless you can find more relevant evidence.

(8) Conclusions are too long and not concise, which need to be modified. I would suggest some contents in conclusions could be incorporated into discussion part.

Technical corrections

(1) Line 119 'A fifth core' should be 'A fourth core'

(2) I would suggest that r2 and p should be showed in the Fig. 9.

---

## Referee Comment (RC2) · Anonymous Referee #2 · 11 May 2017

The manuscript "Carbon mineralization in Laptev Sea and East Siberian Sea shelf and slope sediment" of Brüchert and co-workers describes interesting sediment data from 19 different stations in the East Siberian Sea. The authors measured depth profiles of geochemical data such as DIC concentrations and their stable isotope signatures, oxygen, sulfate and ammonium concentrations as well as process data of sulfate reduction and oxygen consumption. Furthermore, they used the profile data of manganese and iron to model manganese and iron reduction rates. Based on DIC stable isotope signatures determines and the fraction of DIC from organic matter mineralization they derived the contribution of marine and terrestrial organic matter to overall organic matter decomposition using a common endmember model. Finally they upscaled their data to the outer Laptev Sea and the outer East Siberian Sea. This is an impressive data set from a region, which is only hardly accessible and of which only few data are available. The

presented data are of great interest for readers of Biogeosciences and worth being published.

However, the presentation and interpretation of the data need substantial improvement. The manuscript is very long and contains substantial reiterations. But more importantly, a part of the manuscript has the classical structure of Introduction, M&M, Results and Discussion but a large fraction of Methods and Results is presented in the Discussion only. Some results are even discussed before Methods and Results are presented, which makes the manuscript hard to read. I suggest thoroughly rearranging the manuscript according to the classical structure of research manuscripts and shortening the manuscript by removing reiterations.

In particular, the description of the authors approach of partitioning total organic carbon degradation into terrestrial and marine sources needs substantial attention and should be clearly divided into a description of Methods, Results and Discussion (see below). In this context, I missed the carbon concentrations in the sediments and their stable isotope signatures. I assume the authors measured them and they will help to interpret the results of the "modelled"  $\delta$ 13C signatures of respired organic matter.

Most of the data seem being related to anoxic carbon degradation processes, excluding aerobic organic matter turnover, although oxic processes are responsible for most of the organic carbon decomposition in the studied sediments. The authors should clarify throughout the whole manuscript if they relate to total, oxic or anoxic carbon decomposition.

The conclusions are very long and mainly a reiteration of the results and the discussion. It should be shortened substantially.

Specific comments:

- L 29: Please give the depth used for integrating
- L 48: These C amounts are stored in soils of permafrost landscapes. In the permafrost
itself only 800 Pg are stored, see Hugelius et al. 2014. Please rephrase.

L 56: "qualitatively different rates" is unclear, please rephrase.

L 146 omit one "dissolved"

L 154-155: This is a reiteration from 2.1; please remove one of the descriptions.

L 176 - 200: Please indicate where the measurements were done (on the ship, in the home lab) and how the samples were transported.

L221-223: Reiteration from 2.1, please remove one of the descriptions.

L256: Please describe how texture was determined.

L 257: The designation of colors throughout the sediment description is not unambiguous. Please use an accepted color system such as Munsels.

L 296: These data are not presented in Table 3 and it seems they are not presented in the manuscript at all.

L 341-342: Please identify the station.

L 352-356: Here the presentation of the carbon concentration data are needed. Furthermore, to better illustrate changes in organic matter reactivity, SRR should also be presented normalized to organic carbon.

L 360: Decrease or increase of DIC? Furthermore, data of Station 50 should be presented in Fig. 4 if they are given in the text.

L 368: Please indicate where the rates of sedimentation are shown.

L 381-387: This are results that should be presented in the Results section.

L 395: This is a discussion of results that were not shown. Please show these results in the Results section before discussing them.

L 418: Table 4
L 488-502: This is a description of methods, which should go to the Methods.

L 498: Please clarify "no or very minor". Was isotope fractionation considered or not?

L 500: What means minor. Please show the formula for the calculations.

L 502-506: This should go to the Results.

L 505: -35.8‰ is a very low stable isotope value for organic matter, even lower than found for terrestrial organic matter in the hinterland of the Laptev Sea. To assess the significance of this value the stable isotope signatures of the bulk organic matter are needed. Is it possible that methane oxidation contributed to DIC? Are methane concentration values available and can they be referred to? L 525- 538: This is a method description and should go to the Methods

L 539 – L542: These data should be presented in the results.

L 545 – 570: This paragraph again contains mostly a description of methods. The description was also not completely clear to me. Better present the respective formula used for partitioning degradation rates into terrestrial and marine sources. Furthermore, I understood that the authors only considered organic matter degradation via sulfate reduction. If this is the case, it should be made clear, that this approach gives no information on most of organic matter degradation in the sediments (aerobic processes), which likely consume the most labile fraction of deposited organic matter.

L 571 – 573: I cannot see that the data in Fig. 7 shows the "influence of offshore transport of terrestrial organic matter". The figure rather shows decreasing O2 consumption rates and SRR from the shelf down the slope. Please clarify. Furthermore, the remineralized DIC 13C data from Table 3 rather show increasing terrestrial influence down the slope. How does this relate to the data in Fig. 7?

L 606: outer Siberian shelf sediment

L 609: Please show the carbon data in the Results.
L 622 – 627: This should go to the Methods. Please clarify how the "degradation rate constant of organic matter" (L 620) can be determined by the anaerobic carbon mineralization (L 622) if latter only contributes 4 to 26% to total organic matter turnover in the sediments (Table 4).

L 635: anoxic degradation rate measurements?

L 648-650: This sentence should be rephrased since it is unclear. Which implications? If only anaerobic degradation rates are used in the assessment, isn't it obvious that no information on aerobic decomposition can be derived?

L 654 – 656: These results should go to the Results section.

L 654 - 661: Where are these data shown? Only in text of the discussion?

L 659: This sentence is unclear. Regression line of which data? How do you come from a slope of 5.5 to 18%?

L 663 – 664: The numbers for the contribution of anaerobic organic matter decomposition to total organic matter decomposition are given in Table 4 and are generally lower than 18%. This should be discussed.

L 664 – 666: I cannot follow this conclusion. If the contribution of anaerobic organic matter decomposition is only slightly lower (L 661) this means only that (relatively) more organic matter is degraded aerobically but I do not see any information on "highly reactive marine-derived organic material". Please rephrase.

L 711 – 712: Can these data please be presented in the Results?

L 725 – 728: As I understand this relates only to anaerobic organic carbon mineralization. Please clarify.

L 748 – 151: This sentence is unclear. The manuscript did not present any data on priming. How would priming be assessed by this dataset? How can priming be "deduced from the dual contribution of terrestrial and marine-derived organic matter

BGD
to DIC"? I suggest omitting any reference to priming or show a dataset that relates to priming.

L 1127-1129: Please quote the respective reference.

Table 2: Please give mean values also for SRR and O2 uptake at the East Siberian Shelf and standard deviations for all mean values. Furthermore, indicate why a part of the data are missing.

Table 5: please explain TEAP

Figures 2-5: please give the legend at least in one of the panels.

Fig. 4 + 5: The  $\delta$ -symbol in the axis name is missing

---

## Author Comment (AC1) · 2 Jul 2017

**Answers to anonymous Referee #1**

**Please note that our answers are written to each comment in italic type.**

General comments In the manuscript bg-2017-119 presents some important new data on early diagenetic properties and carbon mineralization in the Laptev and East Siberian Sea shelf and slope sediment. In my opinion, it should eventually be published but requires at least moderate revision. Generally, data set is interesting and solid, and conclusions are believable, but the discussion part is a bit unfocussed and manuscript needs to be more concise. In addition, I would suggest that the effect of physical activities such as resuspention and redeposition on pore-water solute concentrations in some shallow sites should be discussed in this manuscript.

Specific comments

(1) line 207 The boundary conditions of the reaction-transport model should be shown in 2.5.

*Answer: We will provide the boundary conditions for the model runs in a table in the appendix.*

(2) Line 210 Generally, the diffusion process includes molecular diffusion and bioturbation, and is not related to bioirrigation. If the study area has evident bioirrigation process, the bioirrigation term a(C0-C) should be added to the model (a is the irrigation coefficient).

*Answer: We have tested the model fits with and without bioirrigation and found no improved fit to the measured concentration data when bioirrigation was included as a process. The exception to this is only Station 48.*

(3) Table 2 showed that DIC fluxes are apparently lower than oxygen uptake rates , but as far as I know, oxygen uptake rates were similar or lower than DIC fluxes in many estuarine and shelf regions. Are there any important processes for removing the porewater DIC in your study regions? Please explain more about the differences between DIC fluxes and oxygen uptake rates.

*Answer: The most likely explanation for the lower DIC fluxes is probably the difference in methods used for the two measurements. With the 1-cm depth resolution of the porewater analyses we are not capturing the gradient in the topmost cm of sediment, the depth at which under aerobic conditions $CaCO_3$ dissolution may take place that would otherwise explain higher or equal DIC to $O_2$ fluxes. The measured gradient is therefore produced by the low anaerobic carbon mineralization rates below 1 cm sediment depth. Based on major ion analysis of $Ca^{2+}$ and $Mg^{2+}$ we see no evidence of cation removal that may be associated with carbonate precipitation. We therefore argue that the lower DIC than $O_2$ fluxes are a consequence of the different measuring methods that use different vertical resolutions. In support of our assessment, we have now added the DIC fluxes measured for the whole-core incubations that were also used to measured the total $O_2$ uptake with 2D optodes and added these to Table 2. There is very good agreement between the whole core DIC fluxes and the $O_2$ uptake rates, consistent with high mineralization rates in the topmost cm of sediment and minimal $CaCO_3$ dissolution.*

(4) Line 371 Pore-water DIC/NH4+ ratios should be corrected by diffusion and adsorption process or at least the authors should demonstrate that these processes can be neglected.

*Thank you for pointing this out. We will correct our rates by accounting for the difference in diffusion coefficients between $NH_4^+$ and DIC ($HCO_3^-$) and the adsorption coefficient of $NH_4^+$. We would like to point out that this correction has a negligible effect on our data analysis because of the similarity in diffusion coefficients and the small correction for ammonium adsorption.*

(5) Line 639 In general, the net Corg settling rates equal to sum of 210Pb-based Corg mass accumulation rates and oxygen uptake rates if the sedimentary Corg changed little with depth, and thus 210Pb-based Corg mass accumulation rates can be lower or higher than oxygen uptake rates.

*Answer: Thank you. We agree. If samples for Pb-210 dating are taken at 1-cm resolution and the topmost cm is either not sampled cleanly or includes a C-rich flocculen tlayer that is not a large total volume of that depth interval, the true carbon inventory is underestimated and deviations such as the ones we observe arise.*

(6) Line 672 Some shelf stations which were influenced by bioturbation should be written, and these discussions about mixing process look like contrasting with the results of optimal fits of the concentration profiles (line 388). Please explain more about the mixing process.

*Answer: We will make this clearer by adding a table on the model optimization to an appendix, in which we will present the bioturbation and bioirrigation coefficients.*

(7) Line747 It is difficult to build relationship with priming effect based on existing data in this paper unless you can find more relevant evidence.

*Answer: We agree. In response to both reviewers' comments we will remove the treatment of potential priming in the Laptev Sea.*

(8) Conclusions are too long and not concise, which need to be modified. I would suggest some contents in conclusions could be incorporated into discussion part.

*Answer: We will sharpen the conclusions and shorten them, but want to use this section to provide a broader perspective of our study.*

Technical corrections

(1) Line 119 'A fifth core' should be 'A fourth core'
*Answer: The fifth core occurred when duplicate cores were taken for porewater measurements.*

(2) I would suggest that r2 and p should be shown in the Fig. 9.

*Answer: We will include this information.*

**Reply to anonymous Referee #2**

**Please note that our answers are written to each comment in italic type.**

The manuscript "Carbon mineralization in Laptev Sea and East Siberian Sea shelf and slope sediment" of Brüchert and co-workers describes interesting sediment data from 19 different stations in the East Siberian Sea. The authors measured depth profiles of geochemical data such as DIC concentrations and their stable isotope signatures, oxygen, sulfate and ammonium concentrations as well as process data of sulfate reduction and oxygen consumption. Furthermore, they used the profile data of manganese and iron to model manganese and iron reduction rates. Based on DIC stable isotope signatures and the fraction of DIC from organic matter mineralization they derived the contribution of marine and terrestrial organic matter to overall organic matter decomposition using a common endmember model. Finally they upscaled their data to the outer Laptev Sea and the outer East Siberian Sea. This is an impressive data set from a region, which is only hardly accessible and of which only few data are available. The presented data are of great interest for readers of Biogeosciences and worth being published.

However, the presentation and interpretation of the data need substantial improvement. The manuscript is very long and contains substantial reiterations. But more importantly, a part of the manuscript has the classical structure of Introduction, M&M, Results and Discussion, but a large fraction of Methods and Results is presented in the Discussion only.

*Answer: We appreciate the reviewer's view of our manuscript structure. However, the part of the discussion the reviewer refers to includes the development of a mass balance model to assess the marine versus terrestrial organic matter contribution to degradation. As such, the mass balance model approach should not be seen as part of the analytical materials and methods, but as a quantitative discussion of our analytical results including the development of model equations. For example, the derivation of the slope and intercept in the DIC\* versus $\delta^{13}C_{DIC}$ plots is not possible without prior analytical results, and neither is the derivation of the relative contributions of marine and terrestrial matter possible without prior analysis of total fluxes. We feel that splitting this part into the materials and methods section and in the discussion section would actually not reduce, but increase the length of the manuscript without adding clarity and lead to dissected information. We would therefore like to retain the present structure in some parts. However, there are other parts, where we agree with the reviewer and these sections will be moved to results and methods, where appropriate.*

Some results are even discussed before Methods and Results are presented, which makes the manuscript hard to read. I suggest thoroughly rearranging the manuscript according to the classical structure of research manuscripts and shortening the manuscript by removing reiterations.

*Answer: We will try to find all unnecessary repeats and remove them to shorten the manuscript as much as possible.*

In particular, the description of the authors approach of partitioning total organic carbon degradation into terrestrial and marine sources needs substantial attention and should be clearly divided into a description of Methods, Results and Discussion (see below).

*Answer: Please see our comment above. The derivation of the equations shoul be part of the discussion to aid in the flow of the argument in the discussion text, but we will, for examle, move parts of the reaction transport model and the carbon equivalent assignment of anaerobic carbon degradation to the methods and results.*

In this context, I missed the carbon concentrations in the sediments and their stable

isotope signatures. I assume the authors measured them and they will help to interpret the results of the "modelled" _13C signatures of respired organic matter.

*Answer: Yes, we have concentrations and $\delta^{13}C_{org}$ values for some, but not all sediments. We have revised the manuscript to cite these references where data are available (some weren't citeable at the time of the original writing, but are now. Since the manuscript already has many data we chose to focus on the novel rates and porewater chemistry instead. There are already a number of publications of the $C_{org}$ and $\delta^{13}C$ contents of these sediments (e.g., Vonk et al, 2012; Bröder et al., 2016 a,b, Karlsson et al. (2015), Salvado et al., 2016) and we wished not to reiterate similar data that were already published.*

Most of the data seem being related to anoxic carbon degradation processes, excluding aerobic organic matter turnover, although oxic processes are responsible for most of the organic carbon decomposition in the studied sediments. The authors should clarify throughout the whole manuscript if they relate to total, oxic or anoxic carbon decomposition.

*Answer: Although we have been very clear to our opinion in distinguishing the aerobic from the anaerobic degradation processes, we will carefully reevaluate the text where this distinction may be obscure.*

The conclusions are very long and mainly a reiteration of the results and the discussion. It should be shortened substantially.

*Answer: We will carefully evaluate where re-iterations occur and remove them, when necessary. However, we do not agree with the reviewer that our conclusions just reiterate the results. Instead, the conclusions put the discussion into the greater context of the overall Arctic marine carbon cycle and some of the debated questions on the likelihood whether sediment-based terrestrial carbon degradation contributes ocean acidification in this region.*

Specific comments:
L 29: Please give the depth used for integrating

*Answer: 30 cm of sediment; revised*

L 48: These C amounts are stored in soils of permafrost landscapes. In the permafrost itself only 800 Pg are stored, see Hugelius et al. 2014. Please rephrase.

*Answer: Hugelius et al (2014) state that 800 Pg are in perennially frozen permafrost, whereas they state that the estimated SOC storage ranges ranges between 1100 to 1500 Pg. We are referring to the latter number.*

L 56: "qualitatively different rates" is unclear, please rephrase.

*Answer: We agree that qualitative is vague, but this is because the literature is often vague on rate constants. Here the term 'qualitative' refers to the widely used characterization of very reactive as opposed to unreactive organic material without specific reference to a degradation rate constant.*

L 146 omit one "dissolved"

*Done.*

L 154-155: This is a reiteration from 2.1; please remove one of the descriptions.

*Answer: removed "undisturbed sediment surfaces and ..".*

L 176 – 200: Please indicate where the measurements were done (on the ship, in the home lab) and how the samples were transported.

*We have inserted the following sentence: Ammonium was determined on a QUAATRO 4-channel flow injection analyzer (Seal Analytical) on board.  All other porewater analyses were performed at the Department of Geological Sciences, Stockholm University. Samples that were analyzed in the home laboratory remained cold or frozen on board until arrival of the icebreaker Oden in Sweden*

L221-223: Reiteration from 2.1, please remove one of the descriptions.

*Removed one "concentration"*

L256: Please describe how texture was determined.

*Answer: This is a qualitative descriptor of the sediment based on visual inspection. No detailed grain analysis was performed.*

L 257: The designation of colors throughout the sediment description is not unambiguous. Please use an accepted color system such as Munsels.

*Munsell's colour chart was not used in the description. Unfortunately, we do not have sediment cores left to compare these to a colour chart. We have instead used colour descriptions that are commonly used for describing sediment cores from the Ocean Drilling Program.*

L 296: These data are not presented in Table 2 and it seems they are not presented in the manuscript at all.

*We inserted these numbers into Table 2 (modelled/measured) O2 uptake*

L 341-342: Please identify the station.

L 352-356: Here the presentation of the carbon concentration data are needed. Furthermore, to better illustrate changes in organic matter reactivity, SRR should also be presented normalized to organic carbon.

*For information, the $\delta^{13}C$ of these sediments vary between -26 ‰ in the Laptev Sea to -20 in the easternmost Siberian Sea (Salvado et al., 2016, Biogeosciences). The carbon concentrations on the outer shelf vary between 0.4% and 1.5% (Bröder et al., 2016b Org. Geochem, Salvado et al., Biogeosciences, pers. data, unpubl.).*

[Figure]

*We attach here a depth profile of organic carbon (%) and $\delta^{13}C_{org}$ from Station 23 to support our point. There is little variation in the $\delta^{13}C$ data with depth. $C_{org}$ concentrations decrease from about 1.3% to 0.8 %, a 40% decrease. Similar profiles can be found in Bröder et al (2016b, Org. Geochem.). It may be argued that the higher $C_{org}$ concentrations in the topmost cm reflect more degradable organic matter, but the decrease in the actual anaerobic degradation rates with depth (Fig. 3) is much larger, more than a factor 8! Scaling the rates to $C_{org}$ concentrations therefore will show the same trend and imply that the reactivity refers to the bulk org. C, which in fact is not true. We have chosen instead to use the conventional unit of $^{35}S$-sulfate reduction rates per volume sediment (nmol cm$^{-3}$ d$^{-1}$), which allows comparison of these rates to the large global database of marine $^{35}S$-sulfate reduction rates.*

L 360: Decrease or increase of DIC? Furthermore, data of Station 50 should be presented in Fig. 4 if they are given in the text.

*Answer: Decrease was corrected. We meant "increase"*

L 368: Please indicate where the rates of sedimentation are shown.

*Answer: We habe removed 'sedimentation' from the sentence. We infer that sedimentation rates would have decreased abruptly at this depth, because the change in sulfate reduction rates over 1 cm depth cannot be explained by a steady eponentail decrease in organic matter reactivity.*

L 381-387: This are results that should be presented in the Results section.

*Answer: The mixing coefficients derived here are the result of the optimization procedure and a specific result for the different stations and will be moved there. We therefore propose to move the reactive transport modeling results into the results section.*

L 395: This is a discussion of results that were not shown. Please show these results in the Results section before discussing them.

*Answer: See above.*

L 418: Table 4L 488-502: This is a description of methods, which should go to the Methods.

*Answer: We will add a section to the materials and methods on the calculation of the carbon equivalents*

L 498: Please clarify "no or very minor". Was isotope fractionation considered or not?

*Answer: It was not.*

L 500: What means minor. Please show the formula for the calculations.

*Answer: Minor to the extent that it cannot be analytically detected based on $Ca^{2+}$ and $Mg^{2+}$ porewater concentration decreases. The calculation refers to equation (3) in line 492.*

L 502-506: This should go to the Results.

*Answer: We will move this sentence to the results.*

L 505: -35.8‰ is a very low stable isotope value for organic matter, even lower than found for terrestrial organic matter in the hinterland of the Laptev Sea. To assess the significance of this value the stable isotope signatures of the bulk organic matter are needed. Is it possible that methane oxidation contributed to DIC? Are methane concentration values available and can they be referred to?

*Answer: Miller et al. 2016 (Biogeosciences) discuss this issue extensively and have come to the conclusion that methane in general and methane oxidation play no role for the porewater chemistry on the East Siberian slope. We were also surprised by the strong decrease with depth in $\delta^{13}C$ DIC. To check the validity of our results we have simulated depth profiles of DIC assuming degradation of a terrestrial organic matter component with an isotope composition of -28‰. However, such a heavy component requires substantially higher carbon degradation rates to match the decrease in $\delta^{13}C$ of DIC, which does not agree with the rates of oxygen uptake, iron and manganese and sulfate reduction measured here. Further, we considered whether there was loss of $^{13}C$ by precipitation of $CaCO_3$ from porewater. This is, however, also not further substantiated based on $Ca^{2+}$ and $Mg^{2+}$ porewater concentrations and the greater likelihood for $CaCO_3$ dissolution than precipitation under the conditions on the lower slope at 3146 m water depth. Given that there are no good alternative explanations or handling artefacts, we present these values as indicating a strongly $^{13}C$-depleted terrestrial signature, although the bulk $\delta^{13}C_{org}$ for this station (Tesi, unpubl. Data) is around -24 to -25‰.*

L 525- 538: This is a method description and should go to the Methods

*Answer: For the sake of the flow of the discussion, we would like to retain this section in place. It is part of our analytical approach to discussing our data and should not be seen as being strictly part of the analytical section of the raw data.*

L 539 – L542: These data should be presented in the results.

*Answer: We argue for the same reason as above to retain this part.*

L 545 – 570: This paragraph again contains mostly a description of methods. The description was also not completely clear to me. Better present the respective formula used for partitioning degradation rates into terrestrial and marine sources. Furthermore, I understood that the authors only considered organic matter degradation via sulfate reduction. If this is the case, it should be made clear, that this approach gives no information on most of organic

matter degradation in the sediments (aerobic processes), which likely consume the most labile fraction of deposited organic matter.

*Answer: To our knowledge, this is the first time such an approach has been used to extract specific marine and terrestrial carbon fraction degradation rates. The reviewer is correct and this is what we have stated in the text.*

L 571 – 573: I cannot see that the data in Fig. 7 shows the "influence of offshore transport of terrestrial organic matter". The figure rather shows decreasing O2 consumption rates and SRR from the shelf down the slope. Please clarify. Furthermore, the remineralized DIC 13C data from Table 3 rather show increasing terrestrial influence down the slope. How does this relate to the data in Fig. 7?

*Answer: The sentence is intended to provide an explanation for the observed decrease. As discussed earlier in the text, terrestrial organic matter is transported offshore and molecular organic studies have shown that the reactivity of this organic matter decreases offshore, consistent with the observed decrease. The $\delta^{13}C_{DIC}$ data from the slope reflect the carbon degradation processes in deeper, buried sediment. These are apparently to a large extent driven by terrestrial organic matter degradation, albeit at low rates.*

L 606: outer Siberian shelf sediment

*Answer: corrected*

L 609: Please show the carbon data in the Results.

*Answer: Carbon centration data can be found in Bröder et al (2016) Biogeoscience Discussions and Salvado et al (2016) Biogeosciences. We add a citation to the text.*

L 622 – 627: This should go to the Methods. Please clarify how the "degradation rate constant of organic matter" (L 620) can be determined by the anaerobic carbon mineralization (L 622) if latter only contributes 4 to 26% to total organic matter turnover in the sediments (Table 4).

*Answer: We move the analytical approach of this section to the methods section. We are specific that this is the anaerobic degradation rate constant. We also show that the oxygen consumption rates are significantly faster than the anaerobic rates, but our isotope approach cannot resolve the origin of the carbon used for aerobic respiration. We agree with Boetius and Damm (1998) that in principle the marine export is large enough to account for all the oxygen uptake, but this does not disprove that a fraction of terrestrial organic matter is also degraded aerobically. Unfortunately, we cannot resolve this question satisfactorily.*

L 635: anoxic degradation rate measurements?

*Answer: Yes, anaerobic degradation rates is what is meant here.*

L 648-650: This sentence should be rephrased since it is unclear. Which implications? If only anaerobic degradation rates are used in the assessment, isn't it obvious that no information on aerobic decomposition can be derived?

*Answer: We will rephrase this sentence. It is important to point out explicitly that all common field measurements of $O_2$ consumption at the sediment surface cannot distinguish a marine from a terrestrial carbon contribution for a mixed source. We feel this is an important point to make, because degradation of terrestrial-derived carbon can only be achieved with additional*

*experiments using isolated sediment from only the oxygenated layer of the sediment. Given that this layer is only a few millimeters thick, this is not a trivial task and could not be achieved within the framework of this study. Nevertheless, we provide a range of previously unknown baseline data for this region, but want to point that other existing published assessments of terrestrial carbon degradation rates on the Siberian shelf are insufficiently constrained.*

L 654 – 656: These results should go to the Results section.

*Answer: This section will be moved to the results.*

L 654 – 661: Where are these data shown? Only in text of the discussion?

*Answer: This section will be moved to the results.*

L 659: This sentence is unclear. Regression line of which data? How do you come from a slope of 5.6 to 18%?

*Answer: This is the inverse of the slope of 5.6*

L 663 – 664: The numbers for the contribution of anaerobic organic matter decomposition to total organic matter decomposition are given in Table 4 and are generally lower than 18%. This should be discussed.

*Answer: This is the consequence of two methods, (1) the regression analysis and (2) the carbon equivalent apportionnement of the anaerobic degradation processes.*

L 664 – 666: I cannot follow this conclusion. If the contribution of anaerobic organic matter decomposition is only slightly lower (L 661) this means only that (relatively) more organic matter is degraded aerobically but I do not see any information on "highly reactive marine-derived organic material". Please rephrase.

*Answer: We try to clarify this by modifying the sentence. In case there is a misunderstanding, an explanation needs to be found for why there should be a greater proportion of aerobic respiration in the Siberian shelf sediment compared to other shelf sediments. We think a viable explanation is that there is a highly reactive marine fraction in the topmost millimeters of sediment that is not present any longer in the buried sediment, where very unreactive terrestrial organic matter prevails. Most shelf sediments with stronger marine $C_{org}$ contributions in temperate regions would not have such a binomial Corg origin of widely different reactivities and a greater proportion of marine Corg would be buried. Hence, the reducing equivalents produced by anaerobic respiration in the Siberian shelf sediment make up a proportionately smaller fraction of the oxygen consumption compared to other shelf environments.*

L 711 – 712: Can these data please be presented in the Results?

*Answer: Our rates were compared with the figures in Bourgeois et al (2017). These are largely interpolated results or derived from the rates by Boetius and Damm (1998), which are already part of Figure 7 A, B.*

L 725 – 728: As I understand this relates only to anaerobic organic carbon mineralization. Please clarify.

*Yes, this is true. For the reasons that the fine-scale processes in the aerobic zone are not resolvable with the porewater analysis used here, the modelling of the porewaters refers to the anaerobic processes.*

L 748 – 151: This sentence is unclear. The manuscript did not present any data on priming. How would priming be assessed by this dataset? How can priming be "deduced from the dual contribution of terrestrial and marine-derived organic matter to DIC"? I suggest omitting any reference to priming or show a dataset that relates to priming.

*Answer: We will remove the discussion on priming.*

L 1127-1129: Please quote the respective reference.

*Answer: The data are from Canfield et al., 2005; Aquatic Geomicrobiology; we will add this.*

Table 2: Please give mean values also for SRR and O2 uptake at the East Siberian Shelf and standard deviations for all mean values. Furthermore, indicate why a part of the data are missing.

*Answer: In the revised version, this has now been done. In addition, we show values from the total oxygen uptake and for the DIC flux based on thewhole-core flux experiments.*

Table 5: please explain TEAP

*Answer: TEAP are Terminal Electron-Accepting Processes after Stumm and Morgan (2006) Aquatic Chemistry; We have added the full term to the text.*

Figures 2-5: please give the legend at least in one of the panels.

Fig. 4 + 5: The _-symbol in the axis name is missing

*Answer: This is a conversion error from Excel to pdf. We will correct this.*

---

## Author Response (AR2)

**Suggestions for revision**

The revised version of the manuscript "Carbon mineralization in the Laptev and East Siberian Sea shelf and slope sediment" by Brüchert and co-workers was substantially improved in terms of structure and clarity in comparison to the original submission. The authors considered several of my comments but there are still several points that should be addressed before publication.

The assessment of the carbon burial efficiency (section 4.2) is still difficult to follow since the authors assumptions are not clearly described. The burial efficiency of terrestrial organic carbon seems to be derived from total carbon accumulation rates and sulfate reduction rates. But how is this possible if total organic carbon contains both marine and terrestrial carbon? From the authors response to my remarks to the original submission I understand that they assume that only marine organic matter decomposition is responsible for oxygen consumption. This is most likely not the case, as the authors are aware. This is yet a quite long paper and the authors might think about shortening the manuscript by omitting sections that are based on questionable assumptions.

Response: Section 4.2 has now been revised and shortened. We think that our analysis is qualitatively correct, but cannot be sufficiently resolved with respect to the oxic degradation of organic matter. We explain this in the text. Likewise the calculation of degradation constants suffers from our ability to resolve the terrestrial fraction of aerobically degraded material (although we think that this fraction is small compared to the marine fraction). We cannot infer directly how much of the oxygen uptake is due to terrestrial OM degradation. The reviewer's comment refers to the old Boetius and Damm (1998) hypothesis, which is not sufficiently constrained. Based on our $\delta^{13}C_{DIC}$ assessment, we know that terrestrial organic matter is remineralized anaerobically and there is no reason to assume that it is not degraded aerobically. However, the proportion of marine and terrestrial degraded by aerobic and anaerobic degradation likely differs, but we have no direct constraint to tell how much of the $O_2$ uptake is due to terrestrial OM degradation alone, as sorry as we are about this conclusion ourselves.

The discussion still starts with the presentation of new results.
Response: This has been changed.

The numbering of the figures should be according to their reference in the text.
Done

The conclusion may substantially be shortened. Currently it is rather an extension of the discussion than a summary of the main findings of the presented work.
We shortened the text.

Specific comments:
Line 47ff: This statement is still not supported by the cited Hugelius et al. (2014) paper and the statement in the current form is still wrong. The authors should be aware of the difference between permafrost carbon (carbon in permafrost) and carbon in the permafrost region which comprises also carbon stored in non-permafrost deposits. According to Hugelius et al. (2014) 800 Pg C are stored in the permafrost, which is by definition frozen ground for more than 2 years. The phrase "perennially frozen permafrost" in the authors response is a tautology since permafrost is by definition perennially frozen. According to

Hugelius et al. (2014) about 500 Pg is stored in the permafrost region in seasonally or perennially unfrozen material (eg. active layer or river and lake sediments) which is not permafrost. The carbon in this non-permafrost pool contributes to the current carbon cycle and this carbon is the source for the terrestrial carbon in the investigated sediments. If the authors insist to refer to the number of 1100-1500 Pg given in the Hugelius paper they should refer to carbon in the permafrost region, as does the cited paper. The permafrost carbon pool, to which the authors still refer, is 500 Pg smaller, which is a significant number.

Response: We have now adapted the wording recommended by the reviewer and refer to the active layer carbon reservoir, with a remark to the currently frozen reservoir, part of which can thaw and oxidize in the future.

Line 56: Again, please clarify what is meant by "qualitative … rates". A rate is a quantitative measure.

Response: We removed the word qualitative

Line 175: DIC comprises beside dissolved CO2 also bicarbonate and carbonate. Please clarify if only dissolved CO2 was considered or total DIC.

Response: We replaced $CO_2$ with total dissolved inorganic carbon

Line 275: … proportion of mineralized terrestrial…

Response: We changed the wording accordingly.

Line 345: The authors should comment on station 63 where the rates measured with the two methods deviate by a factor of 15.

Response: We discuss this in section 3.4 lines 414-418. Briefly, we observe very low rates of degradation in the buried sediments below 8 cm depth. At the same time, at this station the $O_2$ uptake was very high (>10 mmol $m^{-2}$ $d^{-1}$). Since this is the easternmost station of our study and well under the influence of Pacific-derived nutrient water and given the enriched carbon isotope composition of the total organic matter, the marine carbon contribution here is very strong. It is therefore reasonable to assume that the present-day high $O_2$ uptake is due the mineralization of marine organic matter.

Line 366: The Fe concentration at 11 cm seems above 300µM, please clarify.

Response: The text refers to Station 30, where concentrations are lower and do not reach the concentrations mentioned by the reviewer. This only occurs further the east as is discussed in the following sentence.

Line 458ff: I suggest that these data should be presented in the results section with a reference to the respective figure before discussing them. The discussion should not start with the presentation of new results. Furthermore, the authors should give the statistical tests used for the correlation analysis. Since reduced sulfur oxidation is responsible for only a minor part of oxygen consumption, the authors might think about other reasons for the observed correlation, e.g. burial of labile organic matter as discussed in lines 544-548.

Response:   We have now moved this section to the results section.

Line 463: The slope of the regression line in Fig. 9 referred to in the preceding sentence is 5.5 not 6.1. It is unclear which "data set" is referred to here, since the authors presented a lot of interesting data. Please clarify. The regression analyses discussed here was not presented, and it's difficult to follow the discussion of data that are not presented. Please clearly present data before they are discussed.

Response:   We apologize for the confusion. As stated in the text, non-linear regression of all integrated sulfate reduction rates and oxygen uptake rates gave a slope of 6.1±0.1. The depth-integrated sulfate reduction data are presented in section 3.3, lines 393-397 and Table 2 and then used further in the discussion by comparing them to the $O_2$ uptake rates (Table 2).

Line 513: Please explain DOC and POC

Response:   Changed to dissolved organic carbon and particulate organic carbon

Line 523: Please explain BBL

Response:   Changed to: benthic boundary layer

Lines 563. I still believe that the δ13C values of the bulk organic carbon of the investigated sediments are needed here to give the reader the chance to evaluate the modelled results presented by the authors. It is unclear to me why the authors refuse to present the bulk δ13C results so the reader can compare measured data with the results from the authors model. The authors state that they do not wish to reiterate published data in the discussion but to my opinion, an important aim of the discussion is to compare the presented data in the context of previously published data and the authors do so in other sections of the discussion. Knowing that lowest δ13C-values in the sediments of the investigation sites are at about -26‰ is important to evaluate the presented results. A δ13C-value of -35.8‰ for remineralized DIC is hardly explainable if no methane is involved (as the authors assume) and if the bulk organic carbon has values above -26‰. Hence, the authors might also include a sentence on the limitation of their modelling approach.

Response:   There is a complete list of surface sediment concentrations for the investigated stations, including their isotope composition, and C/N ratios presented in Salvado et al (2017) Biogeosciences. We have added the referenced carbon isotope values to Table 1. A more detailed solid-phase composition data of a larger set of stations will be part of a separate publication.

By revising lines 460-470 (track changes) we have addressed the problem arising from the very light -35.8 value by assessing the range of uncertainty of our method, which is about 3 permil for the data from station 1. In addition, separating the DIC data of Station 1 into two sets – one for the top 20 cm and one for below that depth, yields a DIC endmember of -22.7 for the upper section and the measured light value of -35.8 for the bottom section. The light value of the bottom section possibly reflects the disappearance of the

Lines 582 – 585: It is unclear how a δ13C value of -24‰ for marine organic matter is deduced from the reported Alling et al. (2012) values of -2 to -4 for the DIC. Has Alling et al. (2012) also measured δ13C values for the marine POC and which values did they find? Please clarify.

Response:   In the past weeks a new publication on $\delta^{13}C_{POC}$ in the Laptev and East Siberian Sea was published by Tesi et al (2017) Ocean Science. We have used values reported there for our end members in the Laptev Sea and East Siberian Sea and changed the reference and text of this section.

Lines 620-624: Could the authors please give the rates of Boetius and Damm (1988) here because most readers will not know them by heart.

Response:   These rates are now included in the text and a statistical comparison was made with student's T-test.

Line 633: The authors might use statistical tests to test for significant differences between these datasets.

Response:   We show the results of the student's T-test:

Line 635: East Siberian…

Response:   Corrected

Lines 661 – 663: It is unclear to me how the burial efficiency of terrestrial organic carbon can be estimated by using the accumulation rates of total organic carbon comprised of both terrestrial and marine organic carbon. Please clarify. Furthermore, please explain why only sulfate-reduction was considered and not total anoxic degradation rates.

Lines 664 – 667: This assumption of the approach seems to be refuted by the oxygen consumption rates (see discussions in lines 688 – 694). Please clarify.

Line 667 – 668: Please explain why these numbers are substantially different than in the original manuscript (91±6% and 94±4% in contrast to 69±28% and 79±6% in the original submission).

Response to last three comments:   A significant fraction of iron and manganese reduction is directly coupled to reoxidation of sulfide produced by bacterial sulfate reduction. Simply adding the Fe and Mn reduction to the SRR gives a maximum estimate of anoxic carbon degradation. We have no good handle how much of the metal reduction is heterotrophic and how much is due to reoxidation of sulfide and can therefore only provide minimum and maximum estimates, respectively. Secondly, the degradation process rates all produce $CO_2$, but the isotope composition of the two reflects the mixture of oxidized terrestrial and marine organic carbon and the mass balance explained before allows one to estimate the proportion of terrestrial carbon degradation contributing to

CO$_2$ via the anaerobic processes. Based on the comments by the reviewer, we have  reconsidered the text. In light of the fact that it is recommended that we shorten the text of the manuscript and since it appears that the current calculations have shortcomings because they can only applied to the anaerobic section of the sediment, we have decided to leave out the section on carbon burial efficiency, even though that we think that our calculations are probably qualitatively correct. Including the calculation of a burial efficiency that would include the oxic part of the sediment would require too many assumptions on the accumulation of the terrestrial and marine organic matter proportions that we cannot resolve satisfyingly. It is therefore best to refrain from these calculations.

The revised calculation of a burial efficiency only refers to bulk carbon and does not distinguish the marine and terrestrial fraction. Since the Pb-210 Corg accumulation underestimates the true Corg accumulation, we are now adding the O2 uptake to the Pb-210 Corg accumulation and calculate the burial efficiency from this sum. With this revised calculation, significantly lower values for burial efficiency result.

Line 731: The section numbering needs to be revised and there is no section 4.3
Done

Line 759: The numbers given in the manuscript are above 90%.
This whole section is now revised.

1128: Table 4

References:

[revised manuscript text omitted]

---

## Author Response (AR3)

[revised manuscript text omitted]

(2005).